# Learning-related congruent and incongruent changes of excitation and inhibition in distinct cortical areas

Vahid Esmaeili[1]*, Anastasiia Oryshchuk[1], Reza Asri[1], Keita Tamura[1], Georgios Foustoukos[1], Yanqi Liu[1], Romain Guiet[2], Sylvain Crochet[1], Carl C. H. Petersen[1]*

1 Laboratory of Sensory Processing, Brain Mind Institute, Faculty of Life Sciences, École Polytechnique Fédérale de Lausanne (EPFL), Lausanne, Switzerland, 2 Bioimaging and Optics Core Facility, Faculty of Life Sciences, École Polytechnique Fédérale de Lausanne (EPFL), Lausanne, Switzerland

* vahid.esmaeili@epfl.ch (VE); carl.petersen@epfl.ch (CP)

## Abstract

Excitatory and inhibitory neurons in diverse cortical regions are likely to contribute differentially to the transformation of sensory information into goal-directed motor plans. Here, we investigate the relative changes across mouse sensorimotor cortex in the activity of putative excitatory and inhibitory neurons—categorized as regular spiking (RS) or fast spiking (FS) according to their action potential (AP) waveform—comparing before and after learning of a whisker detection task with delayed licking as perceptual report. Surprisingly, we found that the whisker-evoked activity of RS versus FS neurons changed in opposite directions after learning in primary and secondary whisker motor cortices, while it changed similarly in primary and secondary orofacial motor cortices. Our results suggest that changes in the balance of excitation and inhibition in local circuits concurrent with changes in the long-range synaptic inputs in distinct cortical regions might contribute to performance of delayed sensory-to-motor transformation.

## Introduction

Many brain regions are thought to contribute to the performance of goal-directed sensory-to-motor transformations. An increasingly well-defined sensorimotor transformation studied in rodents is the learned association between a whisker sensory input and licking for reward [1–19]. From a cortical perspective considering whisker-dependent tasks requiring licking for perceptual report, sensory processing is prominent in the somatosensory cortices, whereas neuronal activity linked to motor planning during delay periods is primarily found in premotor cortices, and motor commands are more prominent in primary motor cortex [20–23]. We recently showed that in a whisker detection task with delayed licking, the correct execution of the task involves a stereotypical spatiotemporal sequence of whisker deflection-evoked neuronal firing by which sensory cortex appeared to contribute to exciting frontal cortical regions to initiate neuronal delay period activity [22]. Comparing novice and expert mice, we also found that the learning of the task is accompanied by region- and temporal-specific changes in

**Data Availability Statement:** The data used to generate figures that support the findings of this study are freely available in the Open Access CERN

database Zenodo: https://doi.org/10.5281/zenodo.6511622. The Matlab code used to generate figures that support the findings of this study are freely available in the Open Access CERN database Zenodo: https://doi.org/10.5281/zenodo.6511622.

**Funding:** This work was supported by the Swiss National Science Foundation (310030B_166595, 31003A_182010 and CRSII5_177237) (CCHP), the European Research Council (ERC-2011-ADG 293660) (CCHP), European Union's Marie Skłodowska-Curie Actions (665667, 798617) (KT), the Research Foundation for Opto-science and Technology (KT), the Brain Science Foundation (KT), the Japan Society for the Promotion of Sciences (KT), and the Ichiro Kanehara Foundation (KT). The funders had no role in study design, data collection and analysis, decision to publish, or preparation of the manuscript.

**Competing interests:** I have read the journal's policy and the authors of this manuscript have the following competing interests: CCHP serves on the Editorial Board as an Academic Editor of PLOS Biology.

**Abbreviations:** ALM, anterior lateral motor cortex; AP, action potential; ChR2, channelrhodopsin-2; FDR, false discovery rate; FS, fast spiking; LMI, learning modulation index; OMI, opto modulation index; PV, parvalbumin-expressing; RS, regular spiking; STTC, spike time tiling coefficient; tjM1, tongue-jaw primary motor cortex; VGAT, vesicular GABA transporter; wM1, whisker primary motor cortex; wM2, whisker secondary motor cortex; wS1, whisker primary somatosensory cortex; wS2, whisker secondary somatosensory cortex.

cortical activity [22]. These experience-dependent changes in evoked activity likely result from changes in long-range synaptic inputs and changes within local synaptically connected neocortical microcircuits.

Neocortex has regional specializations and a columnar organization divided into layers each containing many classes of neurons varying across diverse features [24–28]. At the most basic level, neocortical neurons can be classified as excitatory (releasing glutamate) or inhibitory (releasing GABA). Many neocortical excitatory neurons send long-range axons projecting to diverse brain regions, whereas most neocortical inhibitory neurons only have local axonal arborizations, thus contributing primarily to the regulation of local microcircuit activity. The balance between excitation and inhibition is likely to have a major impact on neocortical microcircuit computations, and previous work has suggested important changes in this balance across development, brain states, sensorimotor processing and models of brain diseases [29–36]. Inhibitory GABAergic neurons can be further divided into many subclasses, with one of the most prominent being the parvalbumin-expressing (PV) neurons. PV cells provide potent inhibition onto excitatory cells by prominently innervating either the soma and proximal dendrites or the axonal initial segment, thus playing a critical role in controlling the discharge of excitatory neurons. At the millisecond timescale, the PV neurons appear specialized for high-speed synaptic computations with fast membrane time constants and large fast synaptic conductances, receiving substantial excitatory input from many nearby excitatory neurons as well as long-range inputs [37–42]. Within a neocortical microcircuit, PV neurons are likely to play a critical role in controlling the balance between excitation and inhibition. PV cells typically fire at high rates and have short action potential (AP) durations that can be identified from extracellular recordings. In fact, neurons recorded from extracellular recordings are typically classified based of their AP duration, as regular spiking (RS) units, which have broad AP waveforms and correspond mostly to excitatory neurons, and fast spiking (FS) units, which have narrow AP waveforms and largely correspond to inhibitory PV neurons. Previous whisker-related studies have reported experience-dependent plasticity of both excitatory and inhibitory synaptic transmission, with prominent changes reported in PV GABAergic neurons, for example, following whisker deprivation [43,44]. However, it remains unknown how reward-based learning in whisker-dependent tasks might affect the activity of PV neurons, although previous work has revealed prominent changes in PV neuronal activity in mouse motor cortex during learning of a lever press task [45] and in visual cortex during learning of a visual discrimination task [46].

In the present study, we investigate whether the changes observed during the learning of the whisker detection task with delayed licking are associated with a change in the balance between excitation and inhibition. We used our recently published dataset of high-density silicon probe recordings from 6 cortical regions previously identified to be important during this behavior [22] and compared the changes in evoked activity of RS and FS units. Interestingly, we found that upon task learning, RS and FS showed opposite changes in some cortical areas, suggesting important changes in local computation, whereas in other regions, RS and FS changed in parallel suggesting rather an overall shift in the synaptic drive to these areas.

## Results

### Localization and classification of cortical neurons

In this study, we further analyzed a data set of extracellular silicon probe recordings of neuronal spiking activity we published recently [22]. We focused our analyses on 6 key neocortical regions: whisker primary somatosensory cortex (wS1), whisker secondary somatosensory cortex (wS2), whisker primary motor cortex (wM1), whisker secondary motor cortex (wM2),

anterior lateral motor cortex (ALM) and tongue-jaw primary motor cortex (tjM1) (Fig 1A). These regions participate in a whisker detection task with delayed licking to report perceived stimuli [22]. Mice first went through pretraining to the task structure, which included a brief light flash to indicate trial onset followed 2 seconds later by a brief auditory tone to indicate the beginning of the 1-second reporting period, during which the thirsty mice could lick to receive a water reward (Fig 1B). We recorded from 2 separate groups of mice referred to as "Novice" and "Expert" hereafter, while a brief whisker stimulus was introduced 1 second after the visual cue in a randomized half of the trials, and licking in the reporting window was only rewarded in whisker stimulus trials (Fig 1B and 1C). Expert mice were given additional whisker training through which they learned to lick preferentially in trials with a whisker stimulus (Fig 1B and 1C). However, Novice mice had not learned the stimulus–reward contingency and licked equally in trials with and without whisker stimulus [22]. Through anatomical reconstruction of fluorescently labeled electrode tracks and registration to a digital mouse brain atlas, here, we precisely localize units to specific layers and cortical regions annotated in the Allen Mouse Brain Common Coordinate Framework [47] (Figs 1D and S1). The neuronal location was assigned to the recording site with the largest amplitude spike waveform along the shank of the silicon probe (Fig 1E). Neurons in different cortical regions and layers had diverse firing patterns during task performance (Figs 1F and S1). We further distinguished neurons according to the duration of the AP waveform. In both Novice and Expert mice, we found a bimodal distribution of spike duration, which we labeled as FS units (spike duration below 0.26 ms) and RS units (spike duration above 0.34 ms), according to standard nomenclature [48,49] (Figs 1G and S2). Unexpectedly, we found a larger fraction of FS units in sensory areas compared to frontal areas (S2 Fig), which could in part reflect differential distribution of PV neurons [50] and in part might indicate the known sampling bias of extracellular recordings limited to high-firing neurons, whereas sensory cortex typically has rather sparse activity. During task performance, both FS and RS units had a broad range of baseline firing rates (Fig 1H), which appeared to have a near log-normal distribution in both Novice and Expert mice (S3 Fig). In agreement with previous literature, FS units fired at significantly higher rates than RS units in both Expert and Novice mice (Figs 1I and S3).

To investigate the classification of FS and RS units, we conducted a new set of recordings in which we measured the impact of stimulating genetically defined GABAergic neurons in mice expressing channelrhodopsin-2 (ChR2) under the control of the vesicular GABA transporter (VGAT) [51]. Blue light modulated the firing rate of RS and FS neurons in opposite directions, quantified both at the level of population (Fig 1J) and at the level of individual neurons (Fig 1K). Overall, light stimulation increased the firing rate of FS units (blue light off: 4.3 ± 4.9 Hz; blue light on: 33.9 ± 32.7 Hz; 51 units recorded in 4 mice; nonparametric permutation test, $p < 10^{-4}$), whereas it decreased the spike rate of RS units (blue light off: 3.2 ± 3.4 Hz; blue light on: 1.3 ± 7.2 Hz; 130 units recorded in 4 mice; nonparametric permutation test, $p = 0.009$) (Fig 1L). As a second approach, and to avoid network effects of light stimulation [52], we focused only on the first 10-ms window after the onset of light stimulation and identified the opto-tagged neurons based on their fidelity of responses, response onset latency and jitter (Figs 1M–1O and S4). A larger fraction of neurons was opto-tagged among FS neurons compared to RS neurons. These data are therefore consistent with the hypothesis that the majority of FS units are likely to be inhibitory neurons, whereas the majority of RS units are likely to be excitatory neurons.

## Strong task modulation of FS neurons

Many RS units across all 6 cortical regions change their AP firing rates in response to the whisker deflection [22]. Here, we analyzed the responses of FS units during task performance in

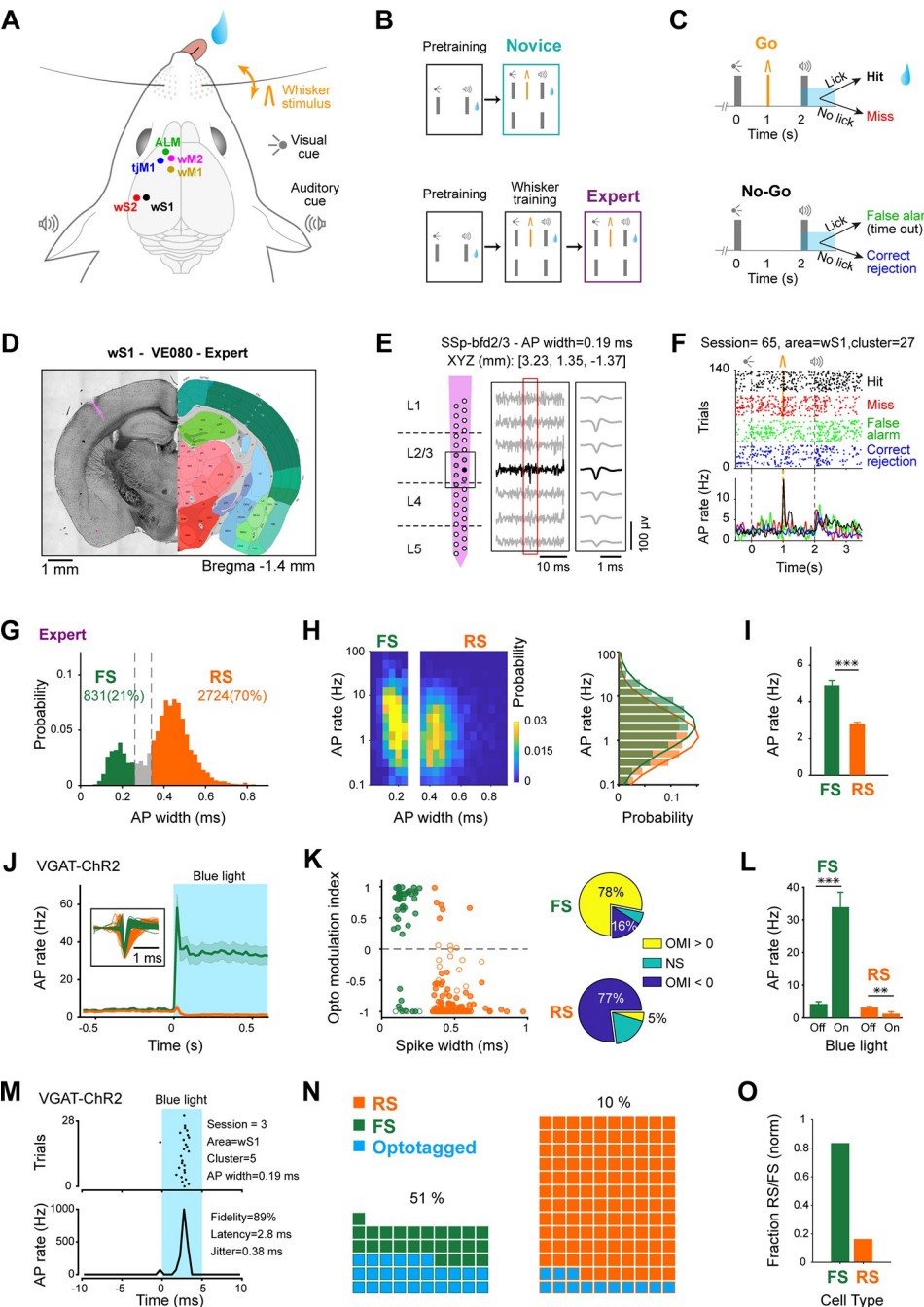

**Fig 1. Multiarea recordings during delayed whisker detection task and assignment of RS and FS units to cortical subdivisions.** (**A**) Schematic of the whisker detection task with delayed response and the targets of silicon probe recordings. (**B**) Training paradigm. Novice and Expert mice were first pretrained in a task, where licking after the auditory cue was rewarded. Expert mice were further trained to only lick in whisker trials. (**C**) Final task structure used during recording sessions (for both groups of mice) and behavioral outcomes. (**D**) Example coronal section of an Expert mouse brain with fluorescent track of a probe in wS1, registered to the Allen Mouse Brain Atlas, https://mouse.brain-map.org [47]. (**E**) Reconstructed laminar location of recording sites of the silicon probe shown in (D) according to the Allen Atlas (left); filtered recorded raw data of 7 sites around one detected spike; and average extracted spike waveform for this example neuron (right). After spike sorting, the position of each cluster (i.e., neuron) was assigned to the location of recording site with the largest spike amplitude (filled circle), and spike width was calculated on the average spike waveform from this site. (**F**) Raster plot and peri-stimulus time histogram for the example neuron shown in (E). Trials are grouped based on outcome. (**G**) Spike width distribution for neurons recorded in Expert mice. Neurons were categorized as FS (spike width <0.26 ms) or RS (spike width >0.34 ms). Neurons with intermediate

spike width (gray bins) were excluded from further analyses. **(H)** Baseline AP rate in Expert mice. Spike width distribution versus baseline AP rate (left) and overlay of spike rate distribution for RS and FS units (right). Note the log-normal distribution of baseline firing rates for both RS and FS units. Normal distributions were fitted to the RS and FS histograms (solid lines). **(I)** Comparison of mean spike rate in RS versus FS neurons of Expert mice. Error bars: SEM. ***: $p < 0.001$, nonparametric permutation test. **(J–O)** Opto-tagging GABAergic neurons in VGAT-ChR2 mice. (J) Grand average firing rate of RS (orange, spike width >0.34 ms, 130 neurons from 4 mice) and FS (green, spike width <0.26 ms, 51 neurons from 4 mice) units upon 100-Hz blue light stimulation (shading shows SEM). Note the suppression of activity in RS and the strong increase of activity in FS population. Inset shows the overlay of average spike waveforms for all RS and FS neurons. (K) OMI versus spike width (*left*) and percentage of modulated neurons (*right*). Each circle represents one neuron, filled circles indicate neurons with significant OMI ($p < 0.05$, nonparametric permutation tests). Pie charts show the percentage of neurons in each group with nonsignificant modulation (NS), and significant positive (OMI > 0) or negative (OMI < 0) modulation upon blue light stimulation. (L) Blue light stimulation in VGAT-ChR2 mice increased the activity of narrow-spike neurons labeled as FS, while it suppressed the activity of broad-spike neurons labeled as RS; 100 to 500 ms after light onset. Error bars: SEM; **: $p < 0.01$; ***: $p < 0.001$. (M) Raster plot and peri-stimulus time histogram during the first 10 ms of the 100-Hz trains of blue light stimulation for an example opto-tagged neuron. (N) Waffle plots showing broad-spike (orange) and narrow-spike (green) neurons, and the opto-tagged neurons (blue) in each group. Numbers indicate the percentage of opto-tagged neurons in each group. (O) Weighted proportion of neurons with narrow (FS) or broad (RS) spike among opto-tagged neurons in (N). The underlying data for Fig 1 can be found in S1 Data. ALM, anterior lateral motor cortex; AP, action potential; FS, fast spiking; OMI, opto modulation index; RS, regular spiking; tjM1, tongue-jaw primary motor cortex; wM1, whisker primary motor cortex; wM2, whisker secondary motor cortex; wS1, whisker primary somatosensory cortex; wS2, whisker secondary somatosensory cortex.

Novice and Expert mice (Fig 2). Averaged across cortical areas and quantified over the first 100 ms after whisker deflection, FS neurons in Novice mice increased their firing rate by 4.6 ± 7.9 Hz (392 units recorded in 8 mice), which was significantly higher (Wilcoxon rank-sum test, $p = 1 \times 10^{-34}$) than the increase in firing rate of RS neurons of 1.0 ± 2.4 Hz (1,089 units recorded in 8 mice) (Fig 2A). Task-modulated RS and FS neurons were mainly excited, with only a small fraction showing significant reduction in firing rate (Fig 2B). Similarly, for Expert mice, whisker deflection evoked an increase of FS firing rate of 4.7 ± 9.1 Hz (831 units recorded in 18 mice) which was significantly higher (Wilcoxon rank-sum test, $p = 4 \times 10^{-71}$) than the increase in firing rate of RS neurons of 1.1 ± 3.9 Hz (2,724 units recorded in 18 mice) (Fig 2C). In addition, for Expert mice, FS neurons were more strongly excited during the delay period compared to RS units (change in firing rate of FS neurons: 1.9 ± 4.8 Hz, 831 units recorded in 18 mice; change in firing rate of RS neurons: 0.7 ± 3.2 Hz, 2,724 units recorded in 18 mice; Wilcoxon rank-sum test, $p = 1 \times 10^{-30}$). In Novice mice, there was little delay period activity in either RS or FS units. The largest fraction of modulated neurons during the delay period were FS units in ALM of Expert mice, which were strongly excited (Fig 2D). Analysis of correct rejection trials in Novice and Expert mice revealed that in the absence of the whisker stimulation neuronal activity remained at baseline levels during the delay period in both RS and FS neurons (S5 Fig). Thus, the overall task selectivity of FS unit activity changed in a similar manner across learning compared to our previous quantification of RS units [22], with FS units having overall larger responses.

## Rapid excitation of FS neurons

Investigating fast sensory processing evoked by the whisker deflection, we found an overall similar sequential recruitment of RS and FS units across cortical areas in both Novice and Expert mice (Figs 3, S6 and S7). The earliest excitation occurred in wS1 and wS2, followed by wM1 and wM2 (Figs 3A and 3B, S6 and S7). In wS1 and wS2, FS neurons responded at significantly shorter latency than RS units in both Novice and Expert mice (Fig 3C), as described later in more detail. Among the other areas, in Novice mice FS neurons responded with shorter latency than RS units in wM2 (FS: 47.7 ± 38.7 ms, 44/57 units in 8 mice; RS: 61.7 ± 40.7 ms, 126/244 units in 18 mice; Wilcoxon rank-sum test, $p = 0.047$, false discovery

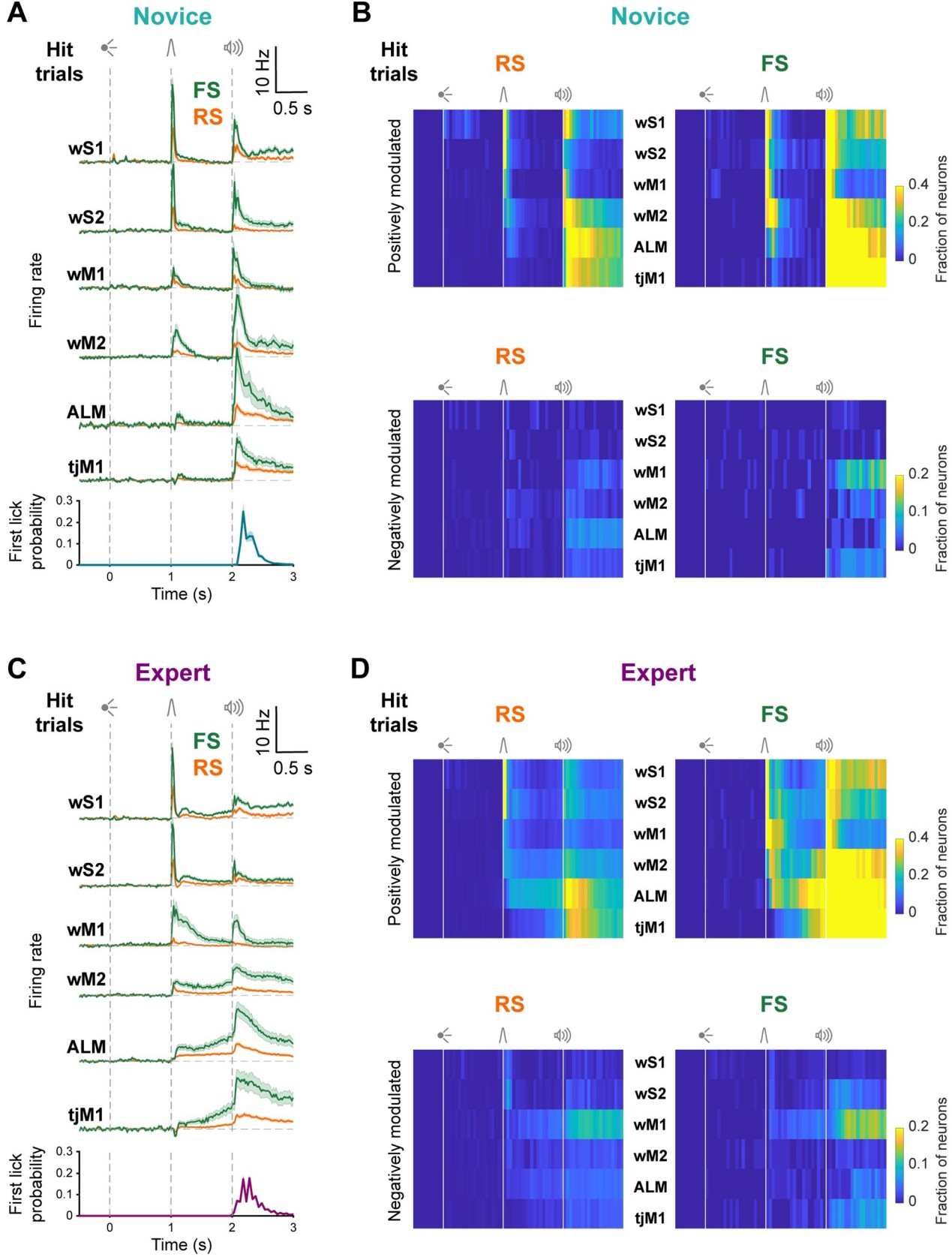

**Fig 2. FS neurons had similar but larger task modulation compared to RS neurons in the same region. (A)** Baseline-subtracted (2 seconds prior to visual onset) population firing rates (mean ± SEM) of RS and FS neurons from different regions of Novice mice are superimposed for hit trials. wS1: 73 RS units in 7 mice, 103 FS units in 7 mice; wS2: 120 RS units in 8 mice, 68 FS units in 8 mice; wM1: 147 RS units in 7 mice, 66 FS units in 7 mice; wM2: 244 RS units in 7 mice, 57 FS units in 7 mice; ALM: 234 RS units in 6 mice, 37 FS units in 5 mice; tjM1: 271 RS units in 8 mice, 61 FS units in 8 mice. Average first lick histogram for all Novice mice is shown in the bottom. **(B)** Percentage of RS (left) and FS (right) neurons in different regions of Novice mice that are positively (top) or negatively (bottom) modulated compared to baseline (nonparametric permutation test, $p < 0.025$). **(C)** Similar to (A), but for Expert mice. wS1: 258 RS units in 15 mice, 237 FS units in 15 mice; wS2: 342 RS units in 12 mice, 161 FS units in 12 mice; wM1: 452 RS units in 11 mice, 134 FS units in 11 mice; wM2: 401 RS units in 10 mice, 107 FS units in 10 mice; ALM: 766 RS units in 12 mice, 109 FS units in 12 mice; tjM1: 505 RS units in 11 mice, 83 FS units in 11 mice. Average first lick histogram for all Expert mice is shown in the bottom. **(D)** Similar to (B), but for Expert mice. Note the difference in color scales for fraction of positively or negatively modulated neurons in b and d. The underlying data for Fig 2 can be found in S2 Data. ALM, anterior lateral motor cortex; FS, fast spiking; RS, regular spiking; tjM1, tongue-jaw primary motor cortex; wM1, whisker primary motor cortex; wM2, whisker secondary motor cortex; wS1, whisker primary somatosensory cortex; wS2, whisker secondary somatosensory cortex.

rate (FDR) corrected for multiple comparison), whereas in Expert mice FS neurons responded with shorter latency than RS units in wM1 (FS: $33.1 ± 35.1$ ms, 101/134 units in 8 mice; RS: $54.2 ± 48.7$ ms, 243/452 units in 18 mice; Wilcoxon rank-sum test, $p = 3 × 10^{-5}$, FDR-corrected for multiple comparison) (Fig 3C). Comparing Novice and Expert mice, the latency of RS units increased in wM1, but decreased in wM2, upon whisker learning (Figs 3D and S6B) [22]. In contrast, FS units did not significantly change their latency across learning in any of the 6 cortical regions (Figs 3D and S6B). These latency differences reveal that task learning is accompanied by fast dynamic changes in the relative timing of the recruitment of FS and RS units across wM1 and wM2.

## Fast sensory processing in wS1 and wS2

Having observed the fastest whisker-evoked responses in wS1 and wS2 (Fig 3), we further compared RS and FS units in these areas, by focusing on their response in the first 50-ms window (Fig 4). The whisker-evoked change in firing rates of RS and FS units in wS1 and wS2 remained unchanged across Novice and Expert mice (Fig 4A–4C). However, in both regions and both groups of mice, FS units had larger evoked responses compared to RS units (Novice wS1: $6.0 ± 9.6$ Hz for 73 RS units versus $13.9 ± 16.1$ Hz for 103 FS units in 8 mice, $p < 10^{-4}$; Novice wS2: $4.0 ± 4.6$ Hz, for 120 RS units versus $12.2 ± 15.8$ Hz for 68 FS units in 8 mice, $p < 10^{-4}$; Expert wS1: $5.3 ± 9.1$ Hz for 258 RS units versus $11.9 ± 13.9$ Hz for 237 FS units in 18 mice, $p < 10^{-4}$; Expert wS2: $4.3 ± 8.7$ Hz for 342 RS units versus $11.5 ± 13.6$ Hz for 161 FS units in 18 mice, $p < 10^{-4}$; nonparametric permutation tests, FDR-corrected for multiple comparison) (Fig 4C). Neuronal responses in wS1 and wS2 often showed a biphasic response; a fast and sharp evoked response followed by a later secondary wave of spiking activity. While, the fast early response remained unchanged (Fig 2A and 2C), the late response increased across learning in RS and FS units of both wS1 and wS2 areas (S9 Fig), consistent with previous work in wS1 in a whisker detection task without a delay period [6].

The latencies of evoked activity in wS1 and wS2 were shorter for FS units compared to RS units for both Novice and Expert mice (Wilcoxon rank-sum tests FDR-corrected for multiple comparison: Novice wS1 $p = 1 × 10^{-7}$; Novice wS2 $p = 1 × 10^{-3}$; Expert wS1 $p = 1 × 10^{-10}$; Expert wS2 $p = 9 × 10^{-6}$) (Fig 4D). Comparing wS1 and wS2 areas, we found no significant difference in RS units response latencies, whereas FS units in wS1 fired at shorter latencies than FS units in wS2 (Wilcoxon rank-sum test FDR-corrected for multiple comparison, Novice: $p = 1 × 10^{-4}$, Expert: $p = 3 × 10^{-4}$). Both wS1 and wS2 therefore responded strongly and similarly to whisker stimulation in both Novice and Expert mice, and no significant change was found in the response of RS or FS units across learning (Fig 4C and 4D).

Optogenetic inactivation by applying blue light in VGAT-ChR2 mice to either wS1 and wS2 during the delivery of the whisker stimulus induced a significant decrease in hit rate [22].

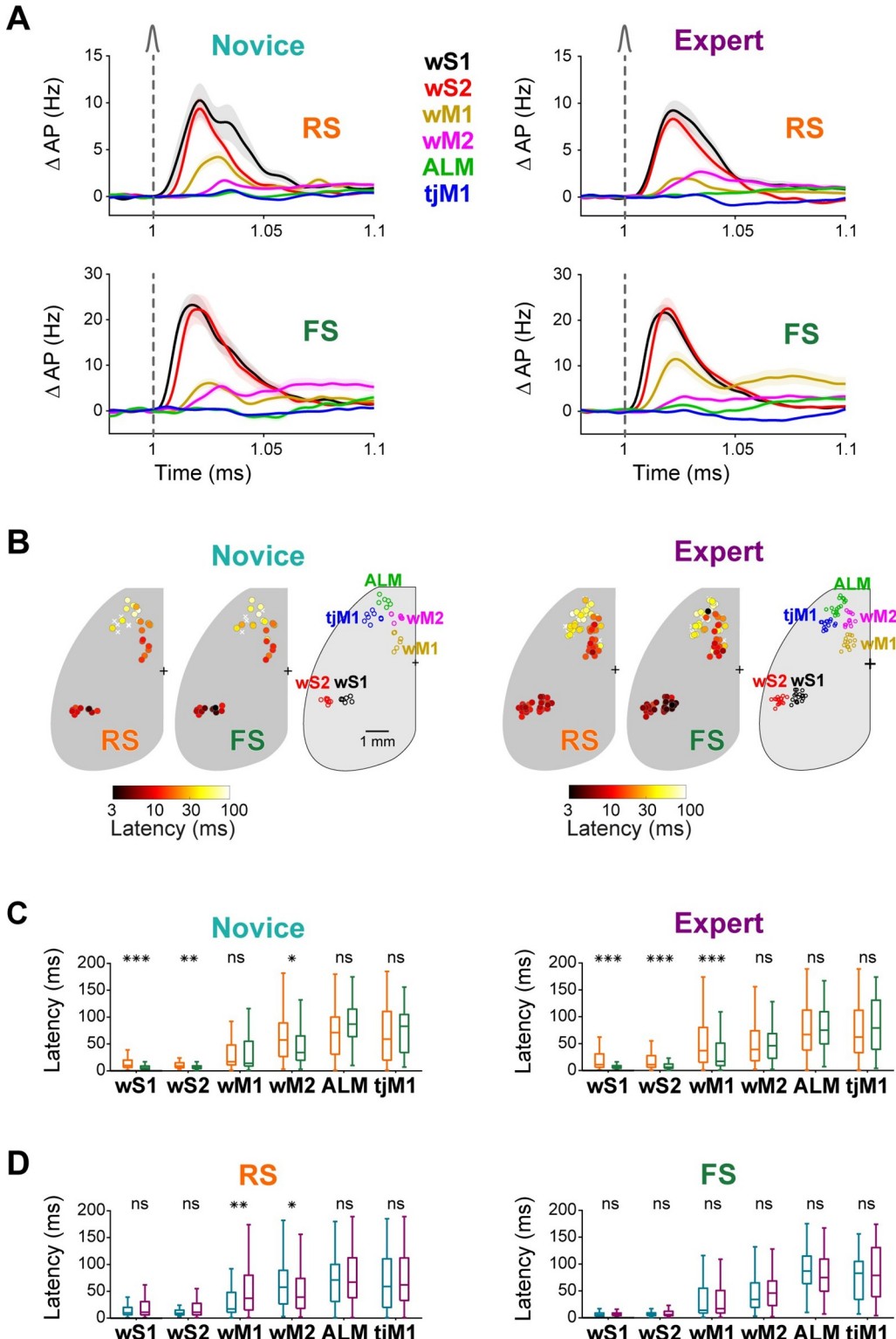

**Fig 3. Fast propagation of sensory responses across cell classes and cortical regions. (A)** Change in firing rate (mean ± SEM) of different cortical regions in the first 100-ms window after whisker deflection for RS (top) and FS (bottom) neurons in Novice (left) and Expert (right) mice (numbers of units and mice are the same as in Fig 2). **(B)** Whisker-evoked response latency maps. For each silicon probe in Novice (left) and Expert (right) mice, average latency of whisker-evoked response is shown separately for RS and FS units. Circles represent silicon probes and are colored

according to the average latency across all responsive neurons recorded on the probe. **(C)** Comparison of latency of RS versus FS neurons in Novice (left) and Expert (right) mice. **(D)** Comparison of latency of neurons from Novice versus Expert mice for RS (left) and FS (right) neurons. In (C) and (D), only neurons with a significant whisker response in the first 200 ms (compared to 200 ms before whisker onset, nonparametric permutation test, $p < 0.05$) were included. Midline represents the median, bottom and top edges show the interquartile range, and whiskers extend to 1.5 times the interquartile range. ***: $p < 0.001$, **: $p < 0.01$, *: $p < 0.05$, ns: $p > = 0.05$. The underlying data for Fig 3 can be found in S4 and S5 Data. ALM, anterior lateral motor cortex; FS, fast spiking; RS, regular spiking; tjM1, tongue-jaw primary motor cortex; wM1, whisker primary motor cortex; wM2, whisker secondary motor cortex; wS1, whisker primary somatosensory cortex; wS2, whisker secondary somatosensory cortex.

Here, we reanalyzed this inactivation data in a direct comparison across these 2 areas and found a significantly stronger deficit induced by inactivation of wS2 compared to wS1 (wS1: Δhit = −0.30 ± 0.13; wS2: Δhit = −0.49 ± 0.12; Wilcoxon signed-rank test, $p = 0.0039$; 9 mice) (Fig 4E). However, potential differences in the spatial extent of the whisker deflection-evoked responses and the efficacy of optogenetic inactivation in wS1 versus wS2 make it difficult to conclude the relative importance of sensory processing in these 2 areas. Nonetheless, the data suggest that neuronal activity in both wS1 and wS2 is involved in execution of this whisker detection task.

## Parallel anatomical pathways from wS1 and wS2 to wM1 and wM2

Neuronal activity in wS1 and wS2 can only contribute to task execution by communicating with other brain regions. Along with various subcortical projections [53,54], innervation of frontal cortical areas might be of particular importance in connecting sensation and movement [10,22]. Neurons in wS1 have previously been shown to innervate wM1 [55–58], but much less is known about the long-range output of wS2. We therefore carried out a set of experiments in which we expressed fluorescent proteins in neurons of wS1 and wS2 to examine their relative innervation targets in frontal cortex (Fig 5A). In the example experiment, we injected virus expressing a red fluorescent protein in wS1 (shown in magenta for better contrast) and a green fluorescent protein in wS2. The fixed brains were imaged through serial section 2-photon tomography and registered to the Allen Mouse Brain Common Coordinate Framework [47] (Fig 5B). As previously shown, wS1 inner-vates frontal cortex with a column of axons in a cortical region we label as wM1 (Fig 5C). Similarly, wS2 axons project to frontal cortex in a columnar manner in a region we label as wM2 (Fig 5D). The location of wM2 appeared to be more anterior compared to the loca-tion of wM1 (Fig 5E and 5F), which is further confirmed by overlaying the projections (Fig 5G). Quantification of the location of the peak on average fluorescence across mice (Fig 5G and 5H, contours) revealed that wM1 was located at 1.0 mm anterior and 1.0 mm lat-eral to bregma, while wM2 was located at 1.9 mm anterior and 1.2 mm lateral to bregma. We further quantified wM1 and wM2 locations by averaging among frontal projection centers from individual mice (Fig 5H, markers) finding similar results (wM1: 1.0 ± 0.1 mm anterior and 1.0 ± 0.1 mm lateral to bregma across 4 mice; wM2: 1.7 ± 0.1 mm ante-rior and 1.0 ± 0.3 mm lateral to bregma across 4 mice). Primary and secondary somatosen-sory cortex therefore map onto frontal cortex in a pattern consistent with mirror-symmetric somatotopy [58] and the frontal projections from visual cortex [59].

## Changes in fast sensory processing in wM1 and wM2

We next investigated the changes in whisker deflection-evoked neuronal activity in wM1 and wM2 across task learning. RS and FS neurons in both wM1 and wM2 and in both Novice and Expert mice showed obvious fast sensory-evoked modulation, dominated by units with

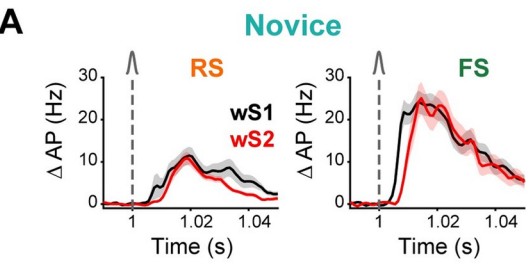

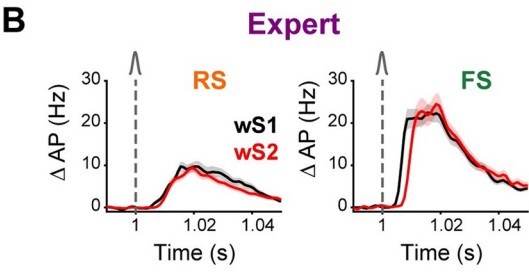

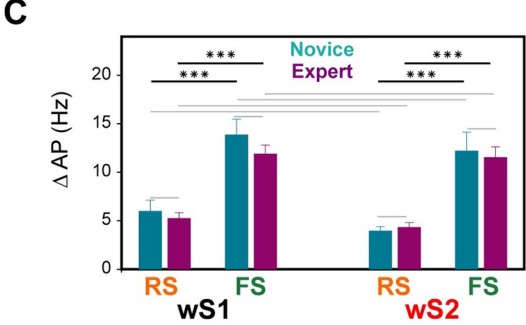

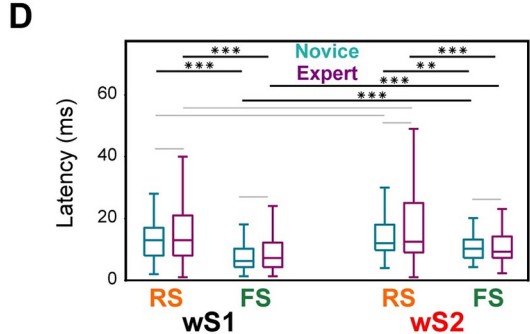

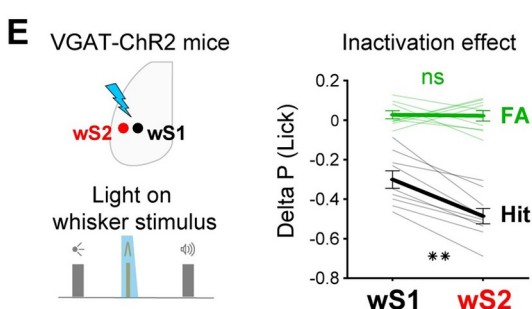

**Fig 4. Fast whisker responses in FS neurons of sensory areas. (A)** Baseline-subtracted (50 ms prior to whisker onset) population firing rate (mean ± SEM) of RS (left) and FS (right) neurons in wS1 and wS2 of Novice mice. wS1: 73 RS units in 7 mice, 103 FS units in 7 mice; wS2: 120 RS units in 8 mice, 68 FS units in 8 mice. **(B)** Same as (A) but for Expert mice. wS1: 258 RS units in 15 mice, 237 FS units in 15 mice; wS2: 342 RS units in 12 mice, 161 FS units in 12 mice. **(C)** Whisker-evoked change in spike rate in the first 50 ms (mean ± SEM) in wS1 and wS2 for RS and FS units and in Novice and Expert mice. ***: $p < 0.001$. Gray lines show nonsignificant comparisons. **(D)** Latency of the whisker-evoked response in wS1 and wS2. Only neurons with a significant whisker response in the first 100 ms (compared to 100 ms before whisker onset, nonparametric permutation test, $p < 0.05$) were included (Novice wS1: 56/73 RS units, 96/103 FS units, 8 mice; Novice wS2: 97/120 RS units, 57/68 FS units, 8 mice; Expert wS1: 190/258 RS units, 210/237 FS units, 18 mice; Expert wS2: 262/342 RS units, 148/161 FS units, 18 mice). Boxplots represent the distribution of the latency defined as the time to reach to half-maximum response. Midline represents the median, bottom and top edges show the interquartile range, and whiskers extend to 1.5 times the interquartile range. ***: $p < 0.001$, **: $p < 0.01$. Gray lines show nonsignificant comparisons. **(E)** Inactivation of wS1 and wS2. Left: Schematic showing the inactivation of wS1 and wS2 areas during whisker stimulus presentation, in VGAT-ChR2 mice [22,51]. Light trials were interleaved with no-light control trials and comprised 1/3 of total trials. Right: Change in hit and FA rate—comparing light and no-light trials—upon optogenetic inactivation of wS1 and wS2. Light colors show individual mice (9 mice), thick lines represent averages, and error bars show SEM. **: $p < 0.01$, ns: $p > = 0.05$. The underlying data for Fig 4 can be found in S6 Data. ChR2, channelrhodopsin-2; FA, false alarm; FS, fast spiking; RS, regular spiking; VGAT, vesicular GABA transporter; wS1, whisker primary somatosensory cortex; wS2, whisker secondary somatosensory cortex.

increased AP firing (Figs 2 and 3 and 6). However, RS and FS neurons changed their activity patterns differentially across learning in these 2 neighboring cortical areas. In wM1, RS units had a smaller whisker-evoked response in Expert compared to Novice mice (Novice: 1.8 ± 3.0 Hz, 147 units recorded in 7 mice, Expert: 0.9 ± 3.9 Hz, 452 units recorded in 11 mice; nonparametric permutation test, $p = 0.002$) (Figs 6A and S10), whereas FS units had a larger response in Expert mice (Novice: 3.1 ± 3.6 Hz, 66 units recorded in 7 mice, Expert: 7.3 ± 16.9 Hz, 134 units recorded in 11 mice; nonparametric permutation test, $p = 0.0008$) (Figs 6B and S10). The ratio of RS to FS firing in wM1 is therefore strongly changed in Expert mice in favor of FS units.

In contrast, we found that neuronal activity in wM2 changed in a very different way across learning compared to wM1. In wM2, whisker deflection evoked an increased AP firing in RS units of Expert mice compared to Novice mice (Novice: 1.0 ± 2.2 Hz, 244 units recorded in 7 mice, Expert: 1.5 ± 4.5 Hz, 401 units recorded in 10 mice; nonparametric permutation test, $p = 0.016$) (Fig 6C and S11), but a decreased firing of FS units (Novice: 4.5 ± 6.8 Hz, 57 units recorded in 7 mice, Expert: 2.7 ± 3.9 Hz, 107 units recorded in 10 mice; nonparametric permutation test, $p = 0.021$) (Figs 6D and S11). The balance of RS to FS unit activity in wM2 is therefore enhanced in favor of RS units across task learning.

To test how the coordination between sensory and motor cortices changed across learning, we quantified interareal interactions between wS1->wM1 and wS2->wM2 in the subset of sessions during which we obtained simultaneous paired recordings from these areas (Fig 6E and 6F). Averaged over individual pairs of neurons, trial-by-trial correlation between evoked activity of wS2-RS units with wM2-RS units increased across learning (Novice: 876 neuron pairs recorded in 6 mice, Expert: 583 neuron pairs recorded in 3 mice; Wilcoxon rank-sum test, $p = 0.039$) while it decreased between wS2-RS units and wM2-FS units (Novice: 343 neuron pairs recorded in 6 mouse, Expert: 209 neuron pairs recorded in 3 mice; Wilcoxon rank-sum test, $p = 2.9 \times 10^{-4}$). Learning-related changes in firing rates might contribute to these apparent changes in correlations. However, while the activity of wM1-FS units increased across learning, the correlation between wS1-RS units and wM1-FS units did not change significantly, nor did the correlation between wS1-RS units and wM1-RS units. As an additional control, we measured interareal pairwise correlations using the spike time tiling coefficient (STTC) method [60], which is suggested to be insensitive to firing rate (S12B Fig). Quantified using STTC analysis, the only significant increase in correlation across learning was observed

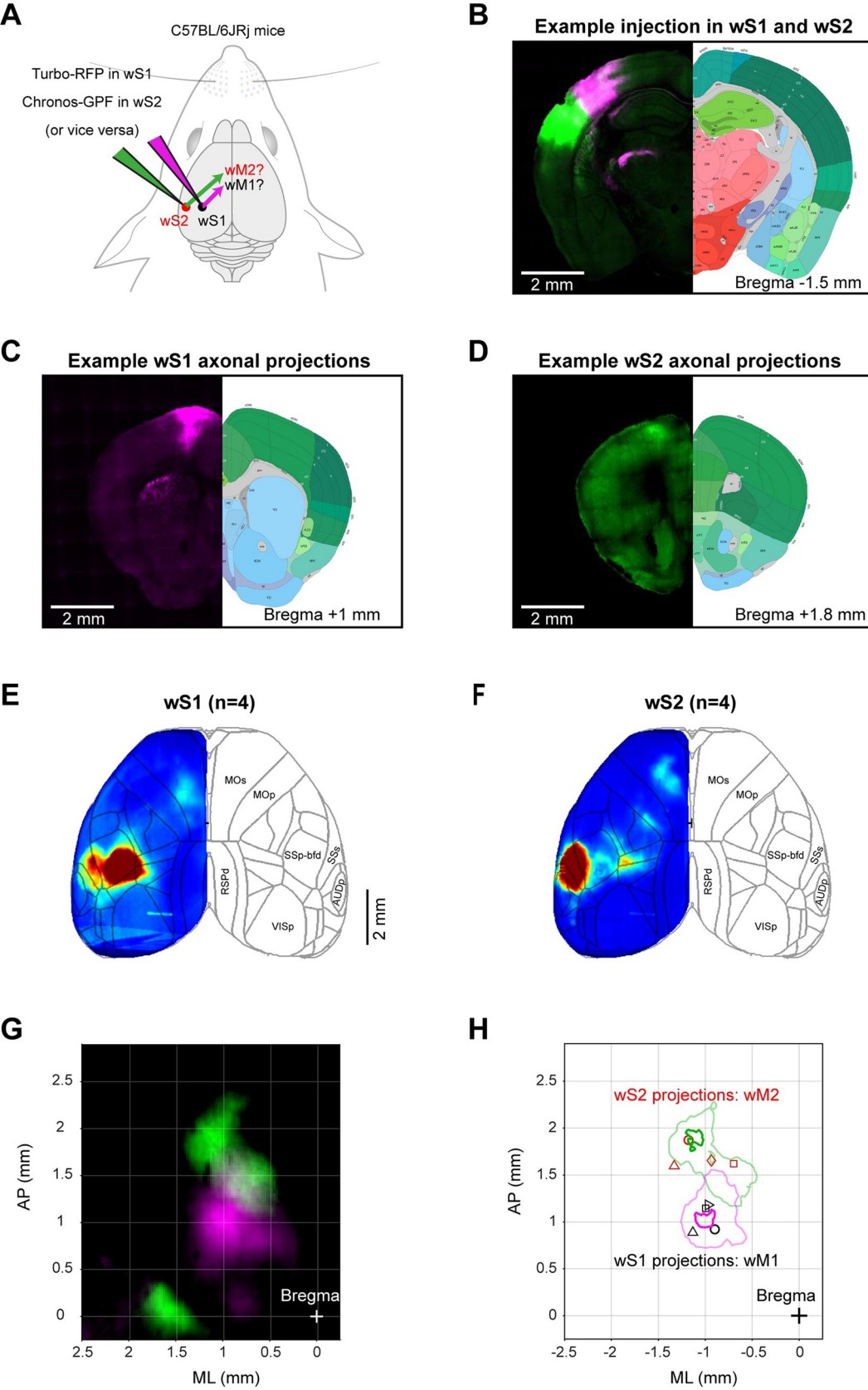

**A** C57BL/6JRj mice

Turbo-RFP in wS1

Chronos-GPF in wS2

(or vice versa)

wM2?
wM1?
wS2 wS1

**B** Example injection in wS1 and wS2

2 mm
Bregma −1.5 mm

**C** Example wS1 axonal projections

2 mm
Bregma +1 mm

**D** Example wS2 axonal projections

2 mm
Bregma +1.8 mm

**E** wS1 (n=4)

MOs
MOp
SSp-bfd
SSs
RSPd
AUDp
VISp
2 mm

**F** wS2 (n=4)

MOs
MOp
SSp-bfd
SSs
RSPd
AUDp
VISp

**G**

AP (mm)
Bregma
ML (mm)

**H**

wS2 projections: wM2
wS1 projections: wM1
Bregma
ML (mm)
AP (mm)

**Fig 5. Distinct frontal projections of wS1 and wS2. (A)** Schematic of anterograde axonal tracing of wS1 and wS2 projections in frontal cortex. Fluorescent proteins of different colors were expressed in wS1 and wS2 regions and frontal projection patterns were identified using anatomical reconstructions and registration to Allen Brain Atlas. **(B–D)** Coronal sections showing example 2-color injections in wS1 (magenta) and wS2 (green) and their frontal projection centers. Viral expression in wS1 and wS2 (B) and frontal sections showing the center of frontal projections in wM1 (C) and wM2 (D). All brains were registered to the Allen Mouse Brain Atlas, https://mouse.brain-map.org. **(E)** Grand average cortical fluorescent map of wS1 projections (4 mice). **(F)** Same as (E) but for wS2 projections (4 mice). **(G)** Overlay of grand average fluorescent map of wS1 (magenta) and wS2 (green) projections in frontal cortex. **(H)** Center of projections from wS1 and wS2 in frontal cortex. Contour plots at 95% and 75% maximum of the grand average fluorescent intensity from wS1 (magenta) and wS2 (green) projections, showing the location of wM1 and wM2, respectively. Markers show the center of projections for different mice. Projections in the same mice are indicated with similar markers. The underlying data for Fig 5 can be found in S7 Data. wM1, whisker primary motor cortex; wM2, whisker secondary motor cortex; wS1, whisker primary somatosensory cortex; wS2, whisker secondary somatosensory cortex.

between wS2-RS and wM2-RS units (Novice: 3,482 neuron pairs recorded in 6 mice, Expert: 2,461 neuron pairs recorded in 3 mice; Wilcoxon rank-sum test, $p = 4.7 \times 10^{-11}$).

Trial-by-trial correlation of the population response showed similar patterns of change across learning in both area pairs as those observed in pair-wise correlation changes (S12A Fig). To further evaluate functional connectivity changes, we identified the number of directional connections (putative direct monosynaptic connections) based on short-latency sharp peaks in the cross-correlograms between pairs of neurons from whisker sensory and whisker motor cortices (Figs 6F and S12C). The percentage of connections between wS2-RS units and wM2-RS units increased significantly across learning (Novice: 3 out of 1,077 pairs in 6 mice, Expert: 17 out of 1,066 pairs in 3 mice; chi-squared proportion test, $p = 0.0032$).

All together, these data suggest that learning might increase the excitation to inhibition ratio of the sensory-evoked response in wM2, but decreases the ratio in wM1 in favor of inhibition. Increased activity of excitatory neurons in wM2 across learning could arise from the increase in functional connectivity between wS2 and wM2 and could, in turn, contribute to driving excitation in other frontal areas including ALM, which is known to be important for the motor planning of licking [20,22].

## Neuronal activity in tongue- and jaw-related motor cortices

Previous studies have identified motor (tjM1) and premotor (ALM) areas of neocortex associated with licking [20,61]. Whisker deflection evoked a rapid decrease in both RS (Fig 7A) and FS (Fig 7B) neuronal activity in tjM1 of Expert mice (RS Novice: 0.0 ± 1.2 Hz, 271 units recorded in 8 mice, RS Expert: −0.6 ± 1.7 Hz, 505 units recorded in 11 mice; nonparametric permutation test, $p < 0.0001$; FS Novice: −0.2 ± 2.1 Hz, 61 units recorded in 8 mice, FS Expert: −1.5 ± 2.6 Hz, 83 units recorded in 11 mice; nonparametric permutation test, $p = 0.0003$). The observed suppression of neuronal activity in tjM1 evoked by the whisker stimulus in Expert mice was present across superficial and deep layers (S13 Fig). The suppression of neuronal activity in tjM1 in Expert mice may help suppress early licking [22]. An active suppression of licking during the response window after the auditory cue is also required in correct rejection trials compared to miss trials, and we previously reported stronger suppression of RS units in correct rejection trials [22]. Here, we similarly observed a larger reduction of activity of FS neurons during the response window in correct rejection trials compared to miss trials (S14 Fig). Thus, in periods when licking should be suppressed, there appears to be a decrease in firing of both RS and FS neurons in tjM1 across learning.

Delay period activity emerges in RS units of ALM after task learning and is causally involved in motor planning [22]. Analysis of FS units revealed a similar activity pattern,

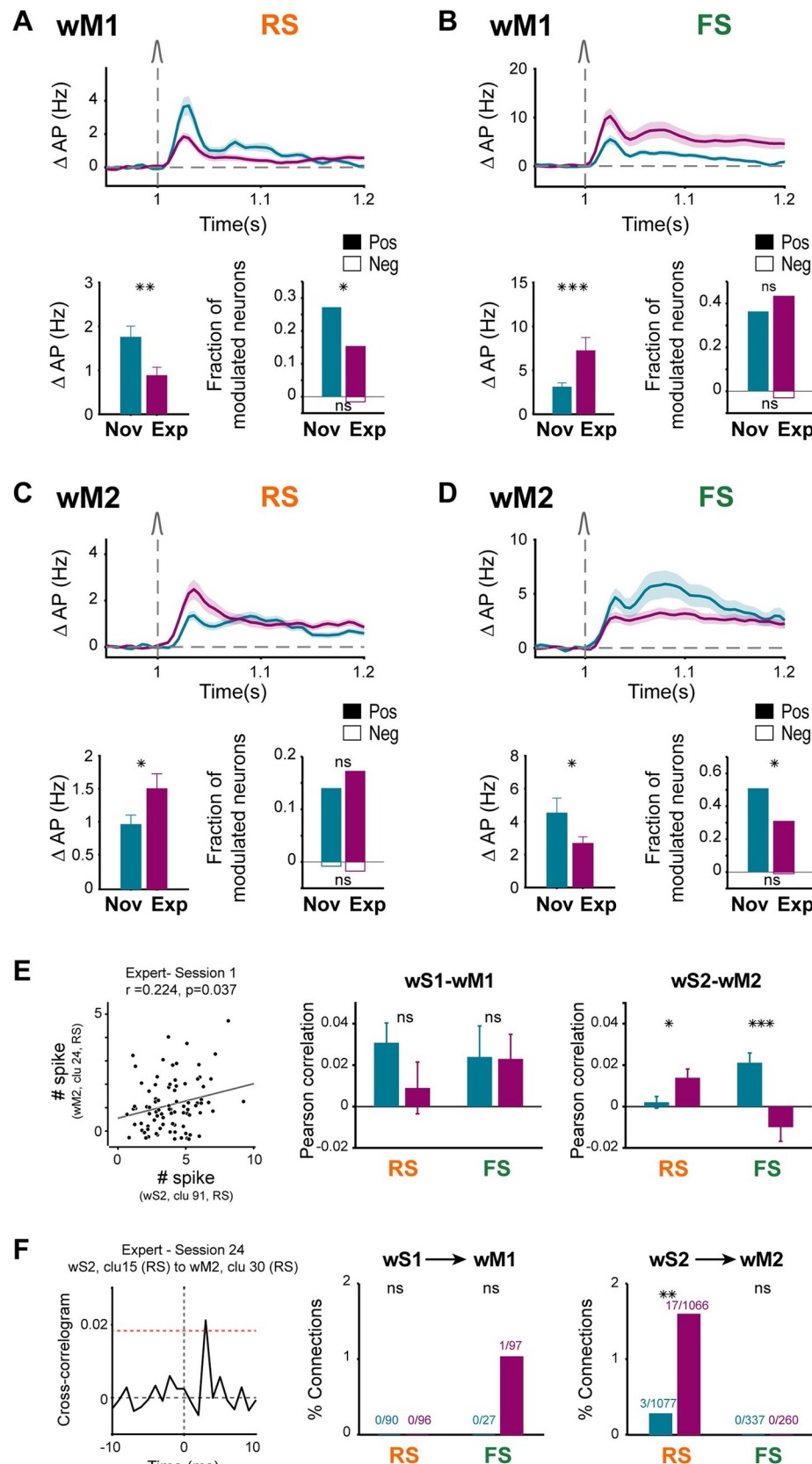

**Fig 6. Learning differently modulated sensory responses of RS and FS neurons in wM1 and wM2 areas. (A)**
Decrease of whisker response in wM1 RS neurons across learning. Top: baseline-subtracted (50 ms prior to whisker
onset) population firing rate (mean ± SEM) overlaid for Novice mice (147 neurons in 7 mice) and Expert mice (452
neurons in 11 mice). Bottom: Comparison of whisker-evoked response in Novice and Expert mice. Bar plots showing
average population rate in 10- to 90-ms window (mean ± SEM) after whisker onset and statistical comparison using
nonparametric permutation test (left) (**: $p < 0.01$; *: $p < 0.05$). The fraction of positively (filled bars) or negatively
(empty bars) modulated neurons in the same window (right). Modulation of individual neurons compared to a similar
window size prior to whisker onset, was identified using nonparametric permutation test ($p < 0.005$). The fractions of
modulated neurons in Novice and Expert were compared using a chi-squared proportion test (*: $p < 0.05$; ns: $p > =$
0.05). **(B)** Increase of whisker response in wM1 FS neurons across learning. Panels are similar to (A) but for wM1 FS
neurons in Novice (66 neurons in 7 mice) and Expert mice (134 neurons in 11 mice) (***: $p < 0.001$). **(C)** Increase of
whisker response in wM2 RS neurons across learning. Panels are similar to (A) but for wM2 RS neurons in Novice
(244 neurons in 7 mice) and Expert mice (401 neurons in 10 mice). **(D)** Decrease of whisker response in wM2 FS
neurons across learning. Panels are similar to (A) but for wM2 FS neurons in Novice (57 neurons in 7 mice) and
Expert mice (107 neurons in 10 mice). **(E)** Pair-wise correlation between sensory and motor cortices in Novice and
Expert mice. Left: Scatter plot showing the trial-by-trial correlation between the whisker-evoked response of an
example pair of neurons in wS2 and wM2. Each circle represents the response of the neuronal pair in one trial. Circles
were jittered slightly for the purpose of visualization. Gray line: least-squares regression. Middle: Average pair-wise
Pearson correlation of wS1-RS units with wM1-RS (110 neuron pairs in 1 Novice mouse, and 68 neuron pairs in 2
Expert mice) and wS1-RS units with wM1-FS units (44 neuron pairs in 1 Novice mouse, and 89 neuron pairs in 2
Expert mice) separately. Right: Average pair-wise Pearson correlation of wS2-RS units with wM2-RS (876 neuron pairs
in 6 Novice mouse, and 583 neuron pairs in 3 Expert mice) and wS2-RS units with wM2-FS units (343 neuron pairs in
6 Novice mouse, and 209 neuron pairs in 3 Expert mice). Error bars: SEM. Statistical comparison between Novice and
Expert was performed using Wilcoxon rank-sum test (ns: $p > = 0.05$; *: $p < 0.05$; ***: $p < 0.001$). **(F)** Interareal
functional connectivity identified based on cross-correlograms. Left: Example cross-correlogram between a pair of
simultaneously recorded neurons from wS2 and wM2. Red dotted line shows the threshold for detecting sharp peaks.
A directional connection from wS2 to wM2 was detected as there is a threshold crossing within the time lags between 0
and 10 ms. Middle: Percentage of detected directional connections from wS1-RS units to wM1-RS and wM1-FS units
in 1 Novice and 2 Expert mice. Right: Percentage of detected directional connections from wS2-RS units to wM2-RS
and wM2-FS units in 6 Novice and 3 Expert mice. The numbers on each bar represent the number of identified
connections and the total number of recorded pairs. The fractions of connections in Novice and Expert were
compared using a chi-squared proportion test (ns: $p > = 0.05$; **: $p < 0.01$). The underlying data for Fig 6 can be found
in S8 Data. FS, fast spiking; RS, regular spiking; wM1, whisker primary motor cortex; wM2, whisker secondary motor
cortex.

indicating that in ALM of Expert mice both RS and FS units increased their firing rate after
whisker stimulus and remain elevated throughout the delay period (RS Novice: 0.1 ± 0.7 Hz,
234 units recorded in 6 mice, RS Expert: 1.4 ± 4.1 Hz, 766 units recorded in 12 mice; nonpara-
metric permutation test, $p = 0.0001$; FS Novice: 0.2 ± 1.5 Hz, 37 units recorded in 5 mice, FS
Expert: 3.7 ± 6.8 Hz, 109 units recorded in 12 mice; nonparametric permutation test,
$p = 0.0001$). Furthermore, in Expert compared to Novice mice, a larger fraction of RS and FS
units were significantly modulated during the delay, primarily with an increase in firing rate
(Fig 7C and 7D). The delay period activity was more prominent in deeper layers of ALM for
both RS and FS neurons (S15 Fig).

Preparatory movements were prominent during delay periods in Expert mice and
accounted for a large part of the neuronal activity during the delay period [22]. Nonetheless,
investigating the subset of quiet trials without delay period movements, we found that signifi-
cant neuronal delay period activity still remains in both RS and FS units (S16 Fig). Therefore,
both RS and FS units in ALM develop persistent delay period activity across learning, which
likely contributes to the storage of a licking motor plan.

## Changes in excitation and inhibition across learning

To the extent that we can equate RS units with excitatory neurons and FS units with inhibitory
neurons (Fig 8A), we can begin to compute changes in the putative balance of excitation and
inhibition as the changes in RS and FS firing rates across learning, providing a simple sum-
mary for comparisons (Fig 8). To do so, for each area (Fig 8B) and cell class (RS or FS), we

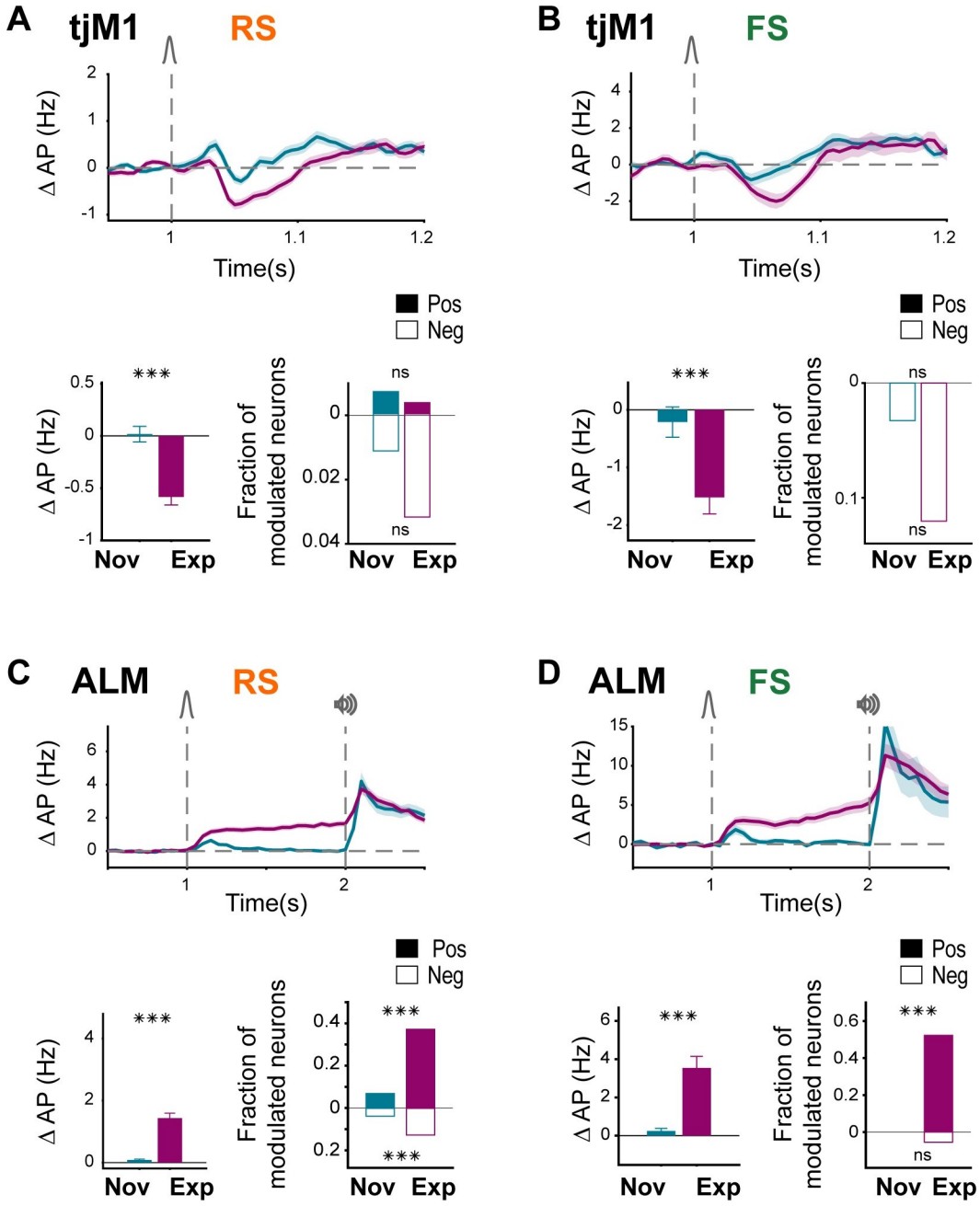

**Fig 7. FS neuronal responses in tjM1 and ALM changed similarly to RS neurons. (A)** Suppression of tjM1 RS neurons in Expert mice. Top: baseline-subtracted (50 ms before whisker onset) firing rate (mean ± SEM) overlaid for Novice (271 RS units in 8 mice) and Expert mice (505 RS units in 11 mice). Bottom: Comparison of whisker-evoked response in Novice and Expert mice. Bar plots showing population rate in 40- to 90-ms window (mean ± SEM) after whisker onset and statistical comparison using nonparametric permutation test (left, ***: $p < 0.001$); fraction of positively (filled bars) or negatively (empty bars) modulated neurons in the same window (right). Modulation of individual neurons compared to a similar window size prior to whisker onset, was identified using nonparametric permutation test ($p < 0.005$). Fraction of modulated neurons in Novice and Expert were compared using a chi-squared proportion test (ns: $p > = 0.05$). **(B)** Suppression of tjM1 FS neurons in Expert mice. Panels are similar to (A) but for tjM1 FS neurons in Novice (61 neurons in 8 mice) and Expert mice (83 neurons in 11 mice). **(C)** Delay activity of RS neurons in Expert mice. Top: baseline-subtracted (1 second before whisker onset) firing rate (mean ± SEM) overlaid for Novice (234 RS units in 6 mice) and Expert mice (766 RS units in 12 mice). Bottom: Comparison of whisker-evoked response in Novice and Expert mice. Bar plots showing population rate in 200- to 1,000-ms window (mean ± SEM) after whisker onset and statistical comparison using nonparametric permutation test (left, ***: $p < 0.001$); fraction of positively (filled bars) or negatively (empty bars) modulated neurons in the same

window (right). Modulation of individual neurons compared to a similar window size prior to whisker onset was identified using nonparametric permutation test ($p < 0.005$). Chi-squared proportion test: ***: $p < 0.001$, ns: $p > = 0.05$. **(D)** Delay activity of ALM FS neurons in Expert mice. Panels are similar to (C) but for ALM FS neurons in Novice (37 FS units in 5 mice) and Expert mice (109 FS units in 12 mice). The underlying data for Fig 7 can be found in S9 Data. ALM, anterior lateral motor cortex; FS, fast spiking; RS, regular spiking; tjM1, tongue-jaw primary motor cortex.

calculated a learning modulation index (LMI) defined as the normalized difference between mean firing rate in Expert and Novice mice. Positive LMI values indicate an increase in neuronal activity across learning, while negative values represent suppression. The putative excitation and inhibition changed in opposite directions in wM1 (RS LMI = −0.33; FS LMI = 0.38) and wM2 (RS LMI = 0.22; FS LMI = −0.25) (Fig 8C and 8D). In contrast, putative excitation and inhibition changed in the same direction across learning in ALM (RS LMI = 0.88; FS LMI = 0.87) and tjM1 (RS LMI = −0.92; FS LMI = −0.75) (Fig 8C and 8D). Subtraction of the LMI of RS from the LMI of FS units as a measure of the change in the putative excitation–inhibition balance across learning, showed a decreased putative excitation–inhibition balance in wM1, but an increased putative excitation–inhibition balance in wM2 (E-I LMI wM1 = −0.72; E-I LMI wM2 = 0.46) (Fig 8E and 8F). Interestingly, the apparent balance of excitation and inhibition thus appears to change differently across learning in distinct cortical areas.

## Discussion

Comparing neuronal activity across task learning revealed distinct changes in RS and FS units in various neocortical areas. Strikingly, in tjM1 and ALM, RS and FS neurons changed firing rates congruently across learning, but in wM1 and wM2, RS and FS changed firing rate incongruently, pointing toward learning-related changes in the balance of cortical excitation and inhibition, with an overall change across learning toward excitation of wM2 and inhibition of wM1.

In wS1 and wS2, we found that there was little change in overall neuronal activity across learning, consistent with a robust coding of the sensory stimulus in these areas of somatosensory cortex (Fig 4). Our results do not rule out a possible reorganization of neuronal activity across learning with some neurons increasing and others decreasing their response to whisker stimulation. Indeed, in a whisker detection task without a delay period, we previously found in wS1 of expert mice that neurons projecting to wS2 had stronger task-related depolarizations compared to neurons projecting to wM1, whereas we found the converse in naive mice [62]. Consistent with an important role for wS2 in whisker detection tasks [19,22,63], here we, found that optogenetic inactivation of wS2, as well as wS1 inactivation, induced a strong impairment in task performance (Fig 4E).

Neuronal activity in wS1 and wS2 can directly influence frontal cortex through direct monosynaptic connections with wM1 and wM2, which we characterized anatomically in this study (Fig 5). Interestingly, neuronal activity in wM1 and wM2 changed profoundly across learning (Fig 6). RS units in wM1 decreased their sensory-evoked response, whereas RS units in wM2 increased their response across learning [22]. Trial-by-trial correlations (Fig 6E) and spike-triggered connectivity analyses (Fig 6F) both pointed to enhanced coupling between wS2-RS units and wM2-RS units, which could, at least in part, result from potentiation of monosynaptic inputs from wS2-RS units to wM2-RS units, although other more complex mechanisms could equally play a role. In contrast, FS units in wM1 increased their response across learning, whereas FS units in wM2 decreased their evoked neuronal activity. Our data thus suggest differential change in the balance between excitation and inhibition with learning in wM1 and wM2, with enhanced sensory-evoked inhibition relative to excitation in wM1 but

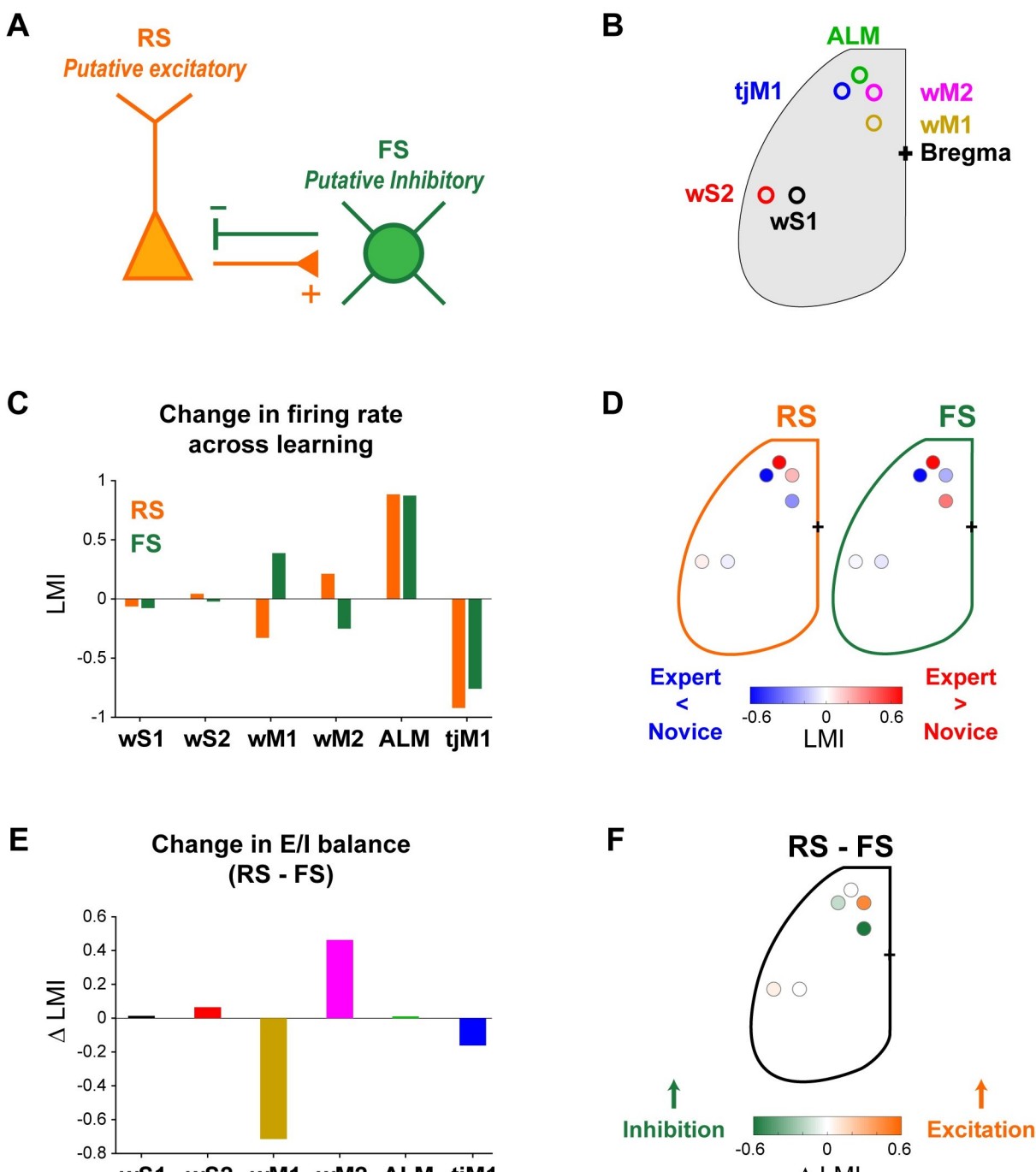

**Fig 8. Diverse changes of putative excitation–inhibition balance in different cortical regions across learning. (A)** FS (putative PV GABAergic inhibitory) neurons and RS (putative glutamatergic excitatory pyramidal) neurons are typically considered to be strongly and reciprocally connected in local cortical microcircuits providing fast balance of excitation and inhibition. **(B)** Schematic showing the location of different cortical regions. **(C)** LMI in different cortical regions for RS and FS units, representing learning-induced change in putative excitation and inhibition, respectively. LMI was quantified as the normalized difference between whisker-evoked firing rate in Novice and Expert mice. **(D)** Map of the putative excitation–inhibition change across learning, shown as LMI across cortical regions for RS and FS neurons. **(E)** Change in the putative excitation–inhibition balance across learning, quantified as difference between LMI of RS and FS neurons in different cortical regions. **(F)** Cortical map of the putative excitation–inhibition balance change across learning, calculated as LMI difference between RS and FS neurons. The underlying data for Fig 8 can be found in S10 Data. ALM, anterior lateral motor cortex; FS, fast spiking; LMI, learning modulation index; PV, parvalbumin-expressing; RS, regular spiking; tjM1, tongue-jaw primary motor cortex; wM1, whisker primary motor cortex; wM2, whisker secondary motor cortex; wS1, whisker primary somatosensory cortex; wS2, whisker secondary somatosensory cortex.

enhanced excitation relative to inhibition in wM2 (Fig 8). Changes in inhibitory neuronal activity could contribute importantly to task learning. Increased recruitment of fast inhibition in wM1 across learning could suppress the response of excitatory neurons in wM1. We speculate that suppression of activity in wM1 could enhance whisker detection performance by reducing whisker movements [64], which otherwise could cause confounding sensory reafference signals. On the other hand, reduced firing of inhibitory neurons in wM2 across learning could allow the excitatory neurons to respond more strongly. Disinhibition of wM2 might be an important step allowing the propagation of whisker sensory information in higher order motor cortex, perhaps contributing to exciting ALM through local intracortical connections [22]. Interestingly, disinhibition of wS1 has previously been reported to contribute to execution of a whisker detection task without a delay [65], suggesting the general importance of considering changes in inhibitory neuronal activity for controlling goal-directed sensorimotor transformations [66–68]. Several mechanisms could contribute to disinhibition, including the activation of GABAergic neurons preferentially innervating other GABAergic neurons, as found in auditory cortex during fear learning [69]. Neuromodulation could also play an important role, for example, through cholinergic reward signals [70], which could drive excitation of vasoactive intestinal peptide-expressing GABAergic neurons [71,72], in turn causing disinhibition through their prominent innervation of PV and somatostatin-expressing GABAergic neurons [73–75].

In contrast to the divergent changes across learning in RS and FS unit activity in wM1 and wM2, RS and FS units changed their activity patterns in the same way in ALM and tjM1 (Fig 7). Suppression of tjM1 activity in Expert mice has a causal role in delayed licking behavior [22]. The rapid suppression of RS units across learning in tjM1 was mirrored by a rapid suppression of FS unit firing (Fig 7A and 7B). Overall, there was thus no apparent change in the balance of excitation and inhibition in tjM1 across learning (Fig 8). The rapid decrease in firing of both RS and FS units evoked by the whisker deflection in Expert mice could result from many different mechanisms, including a possible reduced thalamic or other long-range input to orofacial sensorimotor cortex.

Neuronal delay period activity in ALM is of critical importance for motor planning of licking [20–22]. We found that both RS and FS units increase firing rate during the delay period in Expert mice, but not Novice mice (Fig 7). Similar to tjM1, there was therefore no apparent change in the balance of excitation and inhibition in ALM across learning (Fig 8). Thalamic activity has been shown to be necessary for maintaining ALM activity during delay periods [21,76,77], and increased thalamic input likely excites both RS and FS neurons either directly [41,78–84] or indirectly through local cortical microcircuitry. ALM neurons, in turn, project to thalamic nuclei [21]. In agreement with this, we observed larger delay activity in layer 6 of ALM where many corticothalamic neurons are located [85] (S15 Fig).

In future studies, it will be of importance to better define the various classes of neurons beyond our current classification of RS and FS units. For example, diverse classes of GABAergic neurons can be defined through expression of Cre and Flp recombinase under different promoters [86,87], enabling functional identification of these neurons through opto-tagging [88]. Different classes of excitatory neurons might be best classified through their long-range axonal projections, which could be functionally identified through optogenetic stimulation of axonal branches in target regions [89]. The current study thus takes a first step toward differentiating neuronal activity in various cortical regions across learning, but further experiments will be needed in order to gain a more complete understanding of neocortical cell type–specific changes, as well as, importantly, investigating subcortical regions which are likely to play profound roles in both learning and execution of goal-directed sensorimotor transformations.

## Materials and methods

The results in this study are largely based on further analysis of our recently published dataset available Open Access via the CERN database Zenodo (https://doi.org/10.5281/zenodo.4720013). The methods used to obtain the published dataset were fully described in the accompanying journal publication [22], and are only briefly introduced here. The new analyses are described in detail below. We also carried out 2 new series of experiments: (i) optogenetic tagging of GABAergic neurons; and (ii) anatomical analysis of axonal projections from wS1 and wS2 to frontal cortex. All experimental procedures were approved by the Swiss Federal Veterinary Office (Licences VD1628.7 and VD1889.4) and were conducted in accordance with the Swiss guidelines for the use of research animals. The methods for obtaining the new data are described in detail below. The full data set and analysis code used to generate the figures and results described in this study are available via the Open Access CERN database Zenodo: https://doi.org/10.5281/zenodo.6511622.

### Behavioral paradigm and electrophysiological recordings

Both Novice and Expert mice were trained in the first stage of the task, where in all trials a visual (trial onset) and auditory cues were presented, and licks during a 1-second response window following the auditory cue were rewarded (Fig 1A and 1B). To initiate a trial, mice needed to withhold licking (i.e., not touching the water spout) for a quiet period of 2 to 3 seconds following an inter-trial interval of 6 to 8 seconds. Visual cue (200 ms, green LED) and auditory cue (200 ms, 10 kHz tone of 9 dB added on top of the continuous background white noise of 80 dB) were separated with a delay period that gradually was increased to 2 seconds over Pretraining days. Licking before the response period (Early lick) aborted the trial and introduced a 3- to 5-second timeout. The Expert mice went through a second training phase (Whisker training), in which only Go trials (i.e., trials with a whisker stimulus) were rewarded. Whisker stimulus (10-ms cosine 100 Hz pulse through a glass tube attached to a piezoelectric driver) was delivered to the right C2 whisker 1 second after the visual cue onset in half of the trials. Electrophysiological data from both groups of mice were acquired during the final task conditions (Fig 1C). Novice mice licked in both Go and No-Go trials, while Expert mice had learned to lick selectively in Go trials [22].

Extracellular recordings were performed using single-shank silicon probes (A1x32-Poly2-10mm-50 s-177, NeuroNexus, Michigan, United States of America) with 32 recording sites covering 775 μm of the cortical depth. In each session, 2 probes were inserted in 2 different brain targets acutely. Probes were coated with DiI (1,1′-Dioctadecyl-3,3,3′,3′-Tetramethylindo-carbocyanine Perchlorate, Invitrogen, USA) for post hoc recovery of the recording location (see below). The neural data were filtered between 0.3 Hz and 7.5 kHz and amplified using a digital headstage (CerePlex M32, Blackrock Microsystems, Utah, USA). The headstage digitized the data with a sampling frequency of 30 kHz. The digitized signal was transferred to our data acquisition system (CerePlex Direct, Blackrock Microsystems) and stored on an internal HDD of the host PC for offline analysis.

### Optogenetic tagging of GABAergic neurons

To evaluate to what extent the categorization of units as RS or FS based on spike width is useful for assessing the activity of excitatory versus inhibitory neurons, we performed simultaneous electrophysiological recordings and blue light stimulations in wS1 in 5 sessions from 4 VGAT-ChR2 mice, which express ChR2 in all neocortical GABAergic neuron types. A craniotomy was made over the C2 barrel column, identified by optical intrinsic imaging. For each mouse, a silicon probe (A1x32-Poly2-10mm-50 s-177-A32, NeuroNexus) was slowly lowered to a depth of approximately 1,000 μm in the C2 barrel column, and an optic fiber (400 μm;

NA = 0.39, Thorlabs, USA) coupled to a 470 nm high power LED (M470F3, Thorlabs) was positioned close to the brain surface and the probe. A 100 Hz train of blue light pulses (50% duty cycle, mean power 1 to 2 mW) with the duration of 600 ms was applied. Light pulse train was followed by an additional 100 ms ramping down to prevent rebound excitation. In total, 51 FS and 130 RS units were identified in 5 sessions from 4 mice.

## Anatomical analysis of axonal projections from wS1 and wS2 to frontal cortex

An AAV1.hSyn.TurboRFP.WRPE.rBG (titer: $6.5 \times 10^{13}$ vg/ml, AV-1-PV2642, UPenn Vector Core, USA) was injected at the center of C2 barrel column in wS1 (or in the C2 whisker representation in wS2), and an AAV5.Syn.Chronos-GFP.WPRE.bGH (titer: $3.82 \times 10^{13}$ vg/ml, AV-5-PV3446, UPenn Vector Core) or AAV5.hSyn.hChR2(H134R)-eYFP.WRPE.hGH (titer: $7 \times 10^{12}$ vg/ml, AV-1-26973P, UPenn Vector Core) was injected in the C2 whisker representation in wS2 (or C2 barrel column in wS1). In total, 100 nl of virus was delivered in each area at 300 to 400 μm and 700 to 800 μm below the dura, through a glass pipette (PCR Micropipets 1 to 10 ml, Drummond Scientific, USA) with a 21 to 27 μm inner tip diameter. After 4 weeks of expression, mice were perfused with phosphate buffered saline (PBS) followed by 4% paraformaldehyde (Electron Microscopy Science, USA) in PBS. The brains were postfixed overnight at room temperature. Next, we embedded the brains in 3% to 5% oxidized agarose (Type-I agarose, Merck, Germany) and covalently cross-linked the brain to the agarose by incubating overnight at 4°C in 0.5 to 1% sodium borohydride ($NaBH_4$, Merck) in 0.05 M sodium borate buffer. We imaged the brains in a custom-made 2-photon serial sectioning microscope, which was controlled using MATLAB-based software (ScanImage 2017b, Vidrio Technologies, USA) and BakingTray (https://github.com/BaselLaserMouse/BakingTray, version master: 2019/05/20, extension for serial sectioning) [90]. The setup consists of a 2-photon microscope coupled with a vibratome (VT1000S, Leica, Germany) and a high-precision X/Y/Z stage (X/Y: V-580; Z: L-310, Physik Instrumente, Germany). The thickness of a physical slice was set to be 50 μm for the entire brain and we acquired optical sections at 10 μm using a high-precision piezo objective scanner (PIFOC P-725, Physik Instrumente) in 2 channels (green channel: 500 to 550 nm, ET525/50, Chroma, USA; red channel: 580 to 630 nm, ET605/70, Chroma). Each section was imaged by 7% overlapping $1025 \times 1025$-μm tiles. A 16× water immersion objective lens (LWD 16x/0.80W; MRP07220, Nikon, Japan), with a resolution of 1 μm in X and Y and measured axial point spread function of approximately 5 μm full width at half maximum. After image acquisition, the raw images were stitched using a MATLAB-based software (StitchIt, https://github.com/BaselLaserMouse/StitchIt). The stitched images were then downsampled by a factor of 25 in X and Y obtaining a voxel size of $25 \times 25 \times 25$ μm, using a MATLAB-based software (MaSIV, https://github.com/alexanderbrown/masiv) or using the software Fiji (https://imagej.net/Fiji). The brains were then registered to Allen Mouse Common Coordinate Framework version 3 [47] using a python-based tool (Brainreg, https://github.com/brainglobe/brainreg) [91]. We then acquired 2D maps of cortical projection patterns, by only considering layer 2/3 of cortex and calculating 99% intensity levels across cortical depth using custom-developed analysis routine (https://renkulab.io/projects/guiet.romain/brainreg/files/blob/notebooks/notebooks_napari_brainreg.ipynb). Grand average 2D maps of cortical projections (Fig 5E and 5F) were obtained by first normalizing each mouse's map to its global maximum (i.e., injection site intensity value) and then averaging across mice. The 95% and 75% contours (Fig 5H) for wS1 and wS2 frontal projections sites were calculated on these grand average maps. The center of frontal projection site for individual mice was identified by finding the local maxima in the frontal cortical region (Fig 5H).

## Data analysis and statistics

**Single neuron whisker-evoked response latency.** When measuring the latency of the whisker-evoked response in the firing rate of individual neurons in all cortical areas (Fig 3C and 3D), the analysis was limited to the first 200-ms window following the whisker stimulus. First, we examined whether each neuron was modulated (positively or negatively) in the 200-ms window following the whisker stimulus compared to a 200-ms window prior to the whisker onset. For responsive neurons ($p < 0.05$, nonparametric permutation test), latency—calculated on the temporally smoothed peristimulus time histograms (1-ms nonoverlapping bins filtered with a Gaussian kernel with $\sigma = 10$ ms)—was defined as the time where the neural activity reached half maximum (half minimum for suppressed neurons) within the 200-ms window. Only responsive neurons are included in boxplots in Figs 3C and 3D, and S6. For wS1 and wS2 regions, where neurons had shorter latencies, we recalculated the latencies with higher temporal resolution (Fig 4D). We limited the analysis to 100-ms window following the whisker onset and calculated latencies on smoothed peristimulus time histograms (1-ms non-overlapping bins filtered with a Gaussian kernel with $\sigma = 5$ ms).

**Quantifying opto-tagged neurons.** In recordings from VGAT-ChR2 mice (Figs 1J–1O and S4), we quantified the effect of blue light stimulation on firing rates, on both slow and fast time scales. To quantify the effect of light on each individual neuron we first calculated an opto modulation index (OMI, Fig 1K). OMI was defined for each neuron "n", in light trials, as the normalized difference between the average firing rate during the light window (100 to 500 ms after light onset) versus a baseline of similar duration (−400 to 0 ms prior to light onset):

$$OMI_n = \frac{AP_{light_n} - AP_{baseline_n}}{AP_{light_n} + AP_{baseline_n}}$$

Subsequently, to measure the effect of light stimulation devoid of potential network effects, we focused on the first 10 ms immediately after light onset. We then quantified within this window the following parameters: fidelity, defined as the percentage of trials with at least 1 spike during this window; latency, as the average delay to first spike in trials with at least 1 spike during 10-ms window; and jitter, as the standard deviation of the latency. We then labeled neurons as opto-tagged with fidelity >20%, latency <4.5 ms, and jitter <2 ms (Figs 1M–1O and S4).

**Interareal functional connectivity measures.** Taking advantage of the subset of sessions with simultaneous paired recordings from whisker sensory and motor cortices, we used 2 separate methods to examine the changes across learning in the coordination of interareal neural activity (Figs 6E and 6F and S12). First, we measured Pearson correlation between trial-by-trial whisker evoked responses in pairs of individual neurons recorded from wS1/wS2 (5- to 55-ms window after whisker onset) and wM1/wM2 (10- to 90-ms window after whisker onset) (Fig 6E). For the pair-wise correlation analysis, we only considered neurons with average firing rate > 2.5 Hz within the corresponding analysis windows. Similarly, the Pearson correlation in trial-by-trial average population responses in the same task epochs between pairs of simultaneously recorded areas were quantified (S12A Fig).

As a second measure of pair-wise correlation, which is suggested to be insensitive to firing rate, we applied the STTC approach [60]. The STTC was calculated during a 1-second window centered on the whisker stimulus (S12B Fig) and was defined for spike trains A and B as

$$STTC = \frac{1}{2}\left(\frac{P_A - T_B}{1 - P_A T_B} + \frac{P_B - T_A}{1 - P_B T_A}\right),$$

where $P_A$ and $P_B$ are the proportion of spikes from A falling within $\pm\Delta t$ ($\pm 10$ ms) of a spike in

B and vice versa and $T_A$ and $T_B$ are the proportion of the total recording time that falls within $\pm \Delta t$ of a spike from B or A, respectively.

In addition, we identified directional functional connectivity from wS1 to wM1 and from wS2 to wM2 by calculating cross-correlograms (CCG) during a 1-second window centered on whisker stimulus (Figs 6F and S12C). The CCG was defined as

$$CCG(\tau) = \frac{\frac{1}{M}\sum_{i=1}^{M}\sum_{t=1}^{N}\chi_1^i(t)\chi_2^i(t+\tau)}{\theta(\tau)\sqrt{\lambda_1\lambda_2}},$$

where M is the number of trials, N is the number of bins in the trial, $\chi_1^i$ and $\chi_2^i$ are the spike trains of the 2 units on trial $i$, $\tau$ is the time lag relative to reference spikes, and $\lambda_1$ and $\lambda_2$ are the mean firing rates of the reference and target units, respectively. $\theta(\tau)$ is the triangular function which corrects for the overlap time bins caused by the sliding window [92]. Neurons with firing rate >1 Hz within the analysis window were included in this analysis.

To better capture fast timescale changes related to feedforward connections, cross-correlograms were corrected by subtracting a jittered version [93,94] (S12C Fig):

$$CCG_{corrected} = CCG - CCG_{jittered}$$

The jittered CCG was produced as the average of 100 times resampling the original dataset where spike times within each 25-ms window were randomly permuted across different trials. This method, removes the stimulus-locked and slow timescale correlations larger than the jitter window, while preserving the trial-averaged PSTH and number of spikes for each unit [95]. For each pair of recorded units, the significant directional connection from reference to target neuron was identified if the maximum CCG within time lags between 0 and 10 ms was larger than 6-fold standard deviation of the jitter-corrected CCG flanks (between $\pm$ 50 to 100 ms).

For both analytical methods, in wS1/wS2, we focused only on the RS units, as they are known to have long-range projections. In wM1/wM2, we quantified correlations and directional connections separately for RS and FS units.

**Quantifying LMI.**   The LMI for each cell class ("cc," i.e., RS or FS) and cortical area ("a," i.e., wS1, wS2, wM1, wM2, ALM, or tjM1) was defined as the normalized difference of whisker-evoked response in Novice and Expert mice (Fig 8C and 8D):

$$LMI_{cc,a} = \frac{\Delta AP_{Expert_{cc,a}} - \Delta AP_{Novice_{cc,a}}}{|\Delta AP_{Expert_{cc,a}}| + |\Delta AP_{Novice_{cc,a}}|},$$

where $\Delta AP$ is the grand average change in firing rate (compared to pre-whisker baseline) across all neurons from that mouse group, cortical region and cell class.

**Statistics.**   Data are represented as mean ± SEM unless otherwise noted. The Wilcoxon signed-rank test was used to assess significance in paired comparisons, and the Wilcoxon rank-sum test was used for unpaired comparisons (MATLAB implementations). Analysis of spiking activity was performed using nonparametric permutation test. Comparisons of the number of modulated neurons were performed using a chi-squared proportion test. The statistical tests used and n numbers are reported explicitly in the main text or figure legends. $p$-Values are corrected for multiple comparisons when necessary and methods are indicated in main text or figure legends.

## Supporting information

**S1 Fig. Anatomical localization of neurons. (A)** Magnified example fluorescent track of the silicon probe in wS1 shown in Fig 1D and location of different probe sites after registration to

the Allen Mouse Brain Atlas, https://mouse.brain-map.org. The small rectangular box and black recording site highlight the location of the example neuron shown in (B). **(B)** Silicon probe, example shown in (A), with site locations across cortical layers and example neuron recorded on the probe. Spikes from each neuron were observed across several sites (shown with circles) of the silicon probe. For calculating the spike width for each neuron, the average spike waveform extracted from the recording site with the largest spike peak amplitude (filled circle) was used. Spike width was defined as the time between the spike peak (minimum) to the time voltage came back to baseline level. Gray horizontal line shows spike baseline, and vertical lines mark where spike width was measured. **(C)** Example coronal section of a Novice mouse brain with fluorescent track of a single shank silicon probe in tjM1, registered to the Allen Mouse Brain Atlas, https://mouse.brain-map.org. **(D)** Reconstructed location of different recording sites of the example silicon probe shown in (C) according to Allen Atlas (left), filtered recorded raw data of 7 probe sites around one detected spike, and average extracted spike waveform for this example neuron (right). After spike sorting, the position of each neuron was assigned to the location of recording site across the probe with the largest spike amplitude (filled circle). **(E)** Raster plot and PSTH for the example neuron shown in (D). Trials are grouped and colored based on trial outcome. The underlying data for S1 Fig can be found in S1 Data. PSTH, peri-stimulus time histogram; tjM1, tongue-jaw primary motor cortex; wS1, whisker primary somatosensory cortex.
(TIF)

**S2 Fig. Distribution of spike width in different cortical areas. (A)** Spike width distribution for neurons recorded from Novice mice shown separately for different cortical regions. Neurons were categorized as FS (spike width < 0.26 ms) or RS (spike width > 0.34 ms) in all areas. Neurons with intermediate spike width (gray bars) were excluded from the rest of analysis. Percentage of neurons in each area tagged as RS or FS are shown. **(B)** Same as (A) but for Expert mice. A smaller percentage of FS neurons appears to be found in frontal regions in both Novice (A) and Expert mice (B). The underlying data for S2 Fig can be found in S1 Data. FS, fast spiking; RS, regular spiking.
(TIF)

**S3 Fig. Baseline firing rates of RS and FS neurons. (A)** Spike width distribution versus spike rate for all neurons recorded from Novice mice. **(B)** Spike rate distribution for RS and FS units in Novice (left) and Expert (right) mice. Note the log-normal distribution of baseline firing rates for both RS and FS units in Novice and Expert mice. Normal distributions were fit to the RS and FS histograms (solid lines). **(C)** Comparison of mean spike rate in RS versus FS neurons of Novice and Expert mice. Error bars: SEM. ***: $p < 0.001$, **: $p < 0.01$, *: $p < 0.05$, ns: $p > = 0.05$, nonparametric permutation test, FDR-corrected for multiple comparison. **(D–F)** Same as (A–C), but showing data separately for different cortical areas. The underlying data for S3 Fig can be found in S1 Data. FDR, false discovery rate; FS, fast spiking; RS, regular spiking.
(TIF)

**S4 Fig. Opto-tagging GABAergic neurons in VGAT-ChR2 mice. (A–C)** Criteria for labeling neurons as opto-tagged upon blue light stimulation in VGAT-ChR2 mice. Probability distribution of fidelity scores (A), first spike latency (B), and jitter (C) for RS units (orange, spike width >0.34 ms, 130 neurons from 4 mice) and FS units (green, spike width < 0.26 ms, 51 neurons from 4 mice) measured in the first 10-ms window of blue light stimulation. The thresholds for detection of opto-tagged cells (dotted vertical lines) were defined based on previous literature and dips in the observed probability distributions (i.e., fidelity >20%,

latency <4.5 ms, jitter <2 ms). **(D)** Scatter plot of spike width versus fidelity (left) and distribution of fidelity scores shown with violin plots and bar plots (right). **(E)** Scatter plot of latency versus jitter of light-evoked response for RS and FS units. (**F**) Raster plot and PSTH during the first 10 ms of 100-Hz blue light stimulation for 3 example opto-tagged neurons. The underlying data for S4 Fig can be found in S1 Data. ChR2, channelrhodopsin-2; FS, fast spiking; PSTH, peri-stimulus time histogram; RS, regular spiking; VGAT, vesicular GABA transporter.
(TIF)

**S5 Fig. FS neurons remain at baseline level during correct rejection trials. (A)** Baseline-subtracted (2 seconds prior to visual onset) population firing rates (mean ± SEM) of RS and FS neurons from different regions of Novice mice are superimposed in correct rejection trials. wS1: 73 RS units in 7 mice, 103 FS units in 7 mice; wS2: 120 RS units in 8 mice, 68 FS units in 8 mice; wM1: 147 RS units in 7 mice, 66 FS units in 7 mice; wM2: 244 RS units in 7 mice, 57 FS units in 7 mice; ALM: 234 RS units in 6 mice, 37 FS units in 5 mice; tjM1: 271 RS units in 8 mice, 61 FS units in 8 mice. **(B)** Percentage of RS (left) and FS (right) neurons in different regions of Novice mice that are positively (top) or negatively (bottom) modulated compared to baseline (nonparametric permutation test, $p < 0.05$) in correct rejection trials. **(C)** Similar to (A), but for Expert mice. wS1: 258 RS units in 15 mice, 237 FS units in 15 mice; wS2: 342 RS units in 12 mice, 161 FS units in 12 mice; wM1: 452 RS units in 11 mice, 134 FS units in 11 mice; wM2: 401 RS units in 10 mice, 107 FS units in 10 mice; ALM: 766 RS units in 12 mice, 109 FS units in 12 mice; tjM1: 505 RS units in 11 mice, 83 FS units in 11 mice. **(D)** Similar to (B), but for Expert mice. There appears to be stronger suppression of both RS and FS neurons of tjM1 during the response window in Expert compared to Novice mice. The underlying data for S5 Fig can be found in S3 Data. ALM, anterior lateral motor cortex; FS, fast spiking; RS, regular spiking; tjM1, tongue-jaw primary motor cortex; wM1, whisker primary motor cortex; wM2, whisker secondary motor cortex; wS1, whisker primary somatosensory cortex; wS2, whisker secondary somatosensory cortex.
(TIF)

**S6 Fig. Sequential activation of cortical regions upon whisker stimulation. (A)** Sequential propagation of whisker-evoked spiking responses in hit trials for RS (top) and FS (bottom) neurons from Novice (left) and Expert (right) mice. Mean z-scored firing rate in the first 100-ms window after whisker stimulus is shown. Brain regions are sorted based on their population-average onset latency in RS neurons of Expert mice. **(B)** Cumulative distribution of latency of individual neurons for different cortical areas in RS (top) and FS (bottom) neurons from Novice (left) and Expert (right) mice. Only neurons with significant modulation in the 200-ms window following whisker stimulus compared to a 200-ms window prior to the whisker stimulus are included ($p < 0.05$, nonparametric permutation test). Latency was defined at the half maximum (minimum for suppressed neurons) response within the 200-ms window. The underlying data for S6 Fig can be found in S4 and S5 Data. FS, fast spiking; RS, regular spiking.
(TIF)

**S7 Fig. Early sensory response map across different probes. (A)** Time-lapse maps of whisker-evoked firing rate in RS (top) and FS (bottom) neurons of Novice mice in hit trials. Circles represent different probes and colors show mean baseline-subtracted (50 ms before whisker onset) firing rate across each probe at different time windows. Probes from all Novice mice (43 probes in 8 mice) are superimposed. **(B)** Same as (A) but for Expert mice (90 probes in 18 mice). The underlying data for S7 Fig can be found in S4 and S5 Data. FS, fast spiking; RS,

regular spiking.
(TIF)

**S8 Fig. Faster and larger sensory response in FS neurons across all layers of wS1 and wS2.**
**(A)** Mean z-scored firing rate (left) and baseline-subtracted firing rate (right) of RS (top) and FS (bottom) neurons across different cortical layers of wS1. **(B)** Boxplots representing the distribution of firing rate change (*top*) and response latency (*bottom*) of RS and FS units in different layers of wS1. Only neurons with a significant whisker response in the first 100 ms (compared to 100 ms before whisker onset, nonparametric permutation test, $p < 0.05$) were included. Midline represents the median, bottom and top edges show the interquartile range, and whiskers extend to 1.5 times the interquartile range. *: $p < 0.05$. Gray lines show nonsignificant comparisons. Firing rate change was compared using nonparametric permutation test and latencies were compared using Wilcoxon rank-sum test. **(C, D)** Same as (A, B), but for wS2 neurons. The underlying data for S8 Fig can be found in S6 Data. FS, fast spiking; RS, regular spiking; wS1, whisker primary somatosensory cortex; wS2, whisker secondary somatosensory cortex. (TIF)

**S9 Fig. Layer-specific quantification of RS and FS neuronal responses during the secondary late response in wS1 and wS2 across learning. (A)** Increase of late whisker response in wS1 RS neurons across learning. Left: baseline-subtracted (200 ms prior to whisker onset) population firing rate (mean ± SEM) for all neurons and different cortical layers (L2/3, L5, and L6a) separately overlaid for Novice and Expert mice. The number of neurons is indicated on the figure. Right: change in average spike rate quantified in 150- to 350-ms window after whisker onset relative to similar window size before whisker onset. ***: $p < 0.001$, **: $p < 0.01$, *: $p < 0.05$, ns: $p > = 0.05$, nonparametric permutation test, FDR-corrected for multiple comparison. **(B)** Increase of late whisker response in wS1 FS neurons across learning. Panels are similar to (A) but for wS1 FS neurons in Novice and Expert mice. **(C)** Fraction of wS1 RS neurons across different layers with significant positive (filled bars) or negative (empty bars) modulation late after whisker stimulus (150- to 350-ms window after whisker onset relative to similar window size before whisker onset). Positive or negative modulation of neurons was quantified using nonparametric permutation test ($p < 0.005$). ***: $p < 0.001$, **: $p < 0.01$, *: $p < 0.05$, ns: $p > = 0.05$, chi-squared proportion test. Fractions are reported for groups with more than 5 neurons. **(D)** Similar to (C) but for wS1 FS neurons. **(E–H)** Similar to (A–D) but for wS2. The underlying data for S9 Fig can be found in S6 Data. FDR, false discovery rate; FS, fast spiking; RS, regular spiking; wS1, whisker primary somatosensory cortex; wS2, whisker secondary somatosensory cortex. (TIF)

**S10 Fig. Layer-specific quantification of RS and FS neurons in wM1 across learning. (A)** Decrease of early whisker response in wM1 RS neurons across learning. Left: baseline-subtracted (50 ms prior to whisker onset) population firing rate (mean ± SEM) for different cortical layers (L2/3, L5, and L6a) overlaid for Novice (147 neurons in 7 mice) and Expert (452 neurons in 11 mice) mice. The number of neurons for each layer is indicated on the figure. Right: change in average spike rate quantified in 10- to 90-ms window after whisker onset relative to similar window size before whisker onset. ***: $p < 0.001$, ns: $p > = 0.05$, nonparametric permutation test, FDR-corrected for multiple comparison. **(B)** Increase of whisker response in wM1 FS neurons across learning. Panels are similar to (A) but for wM1 FS neurons in Novice (66 neurons in 7 mice) and Expert (134 neurons in 11 mice) mice. **(C)** Fraction of wM1 RS neurons across different layers with significant positive (filled bars) or negative (empty bars) modulation early after whisker stimulus (10- to 90-ms window after whisker onset relative to

similar window size before whisker onset). Positive or negative modulation of neurons was quantified using nonparametric permutation test ($p < 0.005$). ***: $p < 0.001$, ns: $p > = 0.05$, chi-squared proportion test. Fractions are reported for groups with more than 5 neurons. **(D)** Similar to (C) but wM1 FS neurons. **(E)** Mouse-by-mouse variability and distribution of whisker-evoked response in RS units in wM1 of Novice and Expert mice. (Left) Bar plots showing average firing rate across mice in 10- to 90-ms window (mean ± SEM, 7 Novice and 11 Expert mice) after whisker onset and statistical comparison using nonparametric permutation test (*: $p < 0.05$). Circles show individual mice. (Right) Violin plots showing the distribution of whisker-evoked response in 10- to 90-ms window for all neurons recorded in Novice (147 neurons in 7 mice) and Expert mice (452 neurons in 11 mice). **(F)** Same as (E) but for wM1 FS units in Novice (66 neurons in 7 mice) and Expert mice (134 neurons in 11 mice). The underlying data for S10 Fig can be found in S8 Data. FDR, false discovery rate; FS, fast spiking; RS, regular spiking.
(TIF)

**S11 Fig. Layer-specific quantification of RS and FS neuronal activity in wM2 across learning. (A)** Increase of early whisker response in wM2 RS neurons across learning. Left: baseline-subtracted (50 ms prior to whisker onset) population firing rate (mean ± SEM) for different cortical layers (L2/3, L5, and L6a) overlaid for Novice mice (244 neurons in 7 mice) and Expert mice (401 neurons in 11 mice). The number of neurons for each layer is indicated on the figure. Right: change in average spike rate quantified in 10- to 90-ms window after whisker onset relative to similar window size before whisker onset. *: $p < 0.05$, ns: $p > = 0.05$, nonparametric permutation test, FDR-corrected for multiple comparison. **(B)** Decrease of whisker response in in wM2 FS neurons across learning. Panels are similar to (A) but for wM2 FS neurons in Novice mice (57 neurons in 7 mice) and Expert mice (107 neurons in 10 mice). **(C)** Fraction of wM2 RS neurons across different layers with significant positive (filled bars) or negative (empty bars) modulation early after whisker stimulus (10- to 90-ms window after whisker onset relative to similar window size before whisker onset). Positive or negative modulation of neurons was quantified using nonparametric permutation test ($p < 0.005$). *: $p < 0.05$, ns: $p > = 0.05$, chi-squared proportion test. Fractions are reported for groups with more than 5 neurons. **(D)** Similar to (C) but wM2 FS neurons. **(E)** Mouse-by-mouse variability and distribution of whisker-evoked response in RS units in wM2 of Novice and Expert mice. (Left) Bar plot showing average firing rate across mice in 10- to 90-ms window (mean ± SEM, 7 Novice and 10 Expert mice) after whisker onset and statistical comparison using nonparametric permutation test (*: $p < 0.05$). Circles show individual mice. (Right) Violin plots showing the distribution of whisker-evoked response in 10- to 90-ms window for all neurons recorded in Novice (244 neurons in 7 mice) and Expert mice (401 neurons in 10 mice). **(F)** Same as (E) but for wM2 FS units in Novice (57 neurons in 7 mice) and Expert mice (107 neurons in 10 mice). The underlying data for S11 Fig can be found in S8 Data. FDR, false discovery rate; FS, fast spiking; RS, regular spiking; wM2, whisker secondary motor cortex.
(TIF)

**S12 Fig. Interareal functional connectivity. (A)** Interareal correlation of population response in Novice and Expert mice. (Left) Scatter plot of trial-by-trial average population response between wS1-RS units and wM1-RS units for an example Novice session. Circles were jittered slightly for the purpose of visualization. Gray line: least-squares regression. (Middle) Pearson correlation of trial-by-trial average population response of wS1-RS units versus wM1-RS, and wS1-RS units versus wM1-FS units (1 Novice and 2 Expert mice). (*Right*) Pearson correlation of trial-by-trial population average response of wS2-RS units versus wM2-RS, and wS2-RS units versus wM2-FS units (6 Novice and 3 Expert mice). Circles show individual sessions.

Error bars: SEM. **(B)** Pair-wise correlation between sensory and motor cortices in Novice and Expert mice using the STTC method. *Left*: Average pair-wise STTC correlation of wS1-RS units with wM1-RS (308 neuron pairs in 1 Novice mouse, and 398 neuron pairs in 2 Expert mice) and wS1-RS units with wM1-FS units (112 neuron pairs in 1 Novice mouse, and 139 neuron pairs in 2 Expert mice) separately. *Right*: Average pair-wise Pearson correlation of wS2-RS units with wM2-RS (3,482 neuron pairs in 6 Novice mouse, and 2,461 neuron pairs in 3 Expert mice) and wS2-RS units with wM2-FS units (821 neuron pairs in 6 Novice mouse, and 532 neuron pairs in 3 Expert mice). Error bars: SEM. Statistical comparison between Novice and Expert was performed using Wilcoxon rank-sum test (ns: $p > = 0.05$; *: $p < 0.05$; ***: $p < 0.001$). **(C)** Example cross-correlogram (CCG) from pair of neurons recorded simultaneously in wS2 and wM2 of an Expert mouse with a significant connection; same example pair as shown in Fig 6F, but with CCG from -100 to 100 ms time lags. Jitter correction method (*left*), and detection of significant functional connections (*right*). Significant connections were detected if any threshold crossing happened within 0- to 10-ms time lags (gray bar) of the jitter-corrected CCG. Threshold (red dotted line) was defined as 6-fold standard deviation of the jitter-corrected CCG flanks (red bars). The underlying data for S12 Fig can be found in S8 Data. FS, fast spiking; RS, regular spiking; wM1, whisker primary motor cortex; wM2, whisker secondary motor cortex; wS1, whisker primary somatosensory cortex; wS2, whisker secondary somatosensory cortex.
(TIF)

**S13 Fig. Layer-specific quantification of RS and FS neurons in tjM1 across learning. (A)** Suppression of activity in tjM1 RS neurons across learning. Left: baseline-subtracted (50 ms prior to whisker onset) population firing rate (mean ± SEM) for different cortical layers (L2/3, L5, and L6a) overlaid for Novice mice (271 neurons in 8 mice) and Expert mice (505 neurons in 11 mice). The number of neurons for each layer is indicated on the figure. Right: change in average spike rate quantified in 40- to 90-ms window after whisker onset relative to similar window size before whisker onset. ***: $p < 0.001$, **: $p < 0.01$, *: $p < 0.05$, nonparametric permutation test, FDR-corrected for multiple comparison. **(B)** Suppression of activity in tjM1 FS neurons across learning. Panels are similar to (A) but for tjM1 FS neurons in Novice mice (61 neurons in 8 mice) and Expert mice (83 neurons in 11 mice). **(C)** Fraction of tjM1 RS neurons across different layers with significant positive (filled bars) or negative (empty bars) modulation early after whisker stimulus (40- to 90-ms window after whisker onset relative to similar window size before whisker onset). Positive or negative modulation of neurons was quantified using nonparametric permutation test ($p < 0.005$). ns: $p > = 0.05$, chi-squared proportion test. Fractions are reported for groups with more than 5 neurons. **(D)** Similar to (C) but tjM1 FS neurons. **(E)** Mouse-by-mouse variability and distribution of whisker-evoked response in RS units in tjM1 of Novice and Expert mice. (Left) Bar plots showing average firing rate across mice in 40- to 90-ms window (mean ± SEM, 8 Novice and 11 Expert mice) after whisker onset and statistical comparison using nonparametric permutation test (*: $p < 0.05$). Circles show individual mice. (Right) Violin plots showing the distribution of whisker-evoked response in 40- to 90-ms window for all neurons recorded in Novice (271 neurons in 8 mice) and Expert mice (505 neurons in 11 mice). **(F)** Same as (E) but for tjM1 FS units in Novice (61 neurons in 7 mice) and Expert mice (83 neurons in 11 mice). The underlying data for S13 Fig can be found in S9 Data. FDR, false discovery rate; FS, fast spiking; RS, regular spiking; tjM1, tongue-jaw primary motor cortex.
(TIF)

**S14 Fig. Suppression of activity in tjM1 during response window in no-lick trials of Expert mice. (A)** Stronger suppression of tjM1 RS neurons in correct rejection versus miss trials. Left:

baseline-subtracted (200 ms prior to auditory onset) population firing rate (mean ± SEM) overlaid for correct rejection (blue) and miss trials (red) in Expert mice (505 neurons in 11 mice). Right: change in average spike rate quantified in 200- to 1,000-ms window after auditory onset relative to a 200-ms window prior to auditory onset. *: $p < 0.05$, ns: $p > = 0.05$, nonparametric permutation test. **(B)** Similar to (A) but for FS neurons (83 neurons in 11 mice). **(C)** Similar to (A) but separately for RS neurons of different cortical layers. The number of neurons for each layer is indicated on the figure. *: $p < 0.05$, ns: $p > = 0.05$, nonparametric permutation test, FDR-corrected for multiple comparison. **(D)** Similar to (C) but for FS neurons. **(E)** Fraction of tjM1 RS neurons across different layers with significant positive (filled bars) or negative (empty bars) modulation in correct rejection and miss trials, quantified during response window (200- to 1,000-ms window after auditory onset relative to 200-ms window before auditory onset). Positive or negative modulation of neurons was quantified using nonparametric permutation test ($p < 0.005$). ***: $p < 0.001$, *: $p < 0.05$, ns: $p > = 0.05$, chi-squared proportion test. **(F)** Similar to (E) but tjM1 FS neurons. The underlying data for S14 Fig can be found in S9 Data. FDR, false discovery rate; FS, fast spiking; RS, regular spiking; tjM1, tongue-jaw primary motor cortex.
(TIF)

**S15 Fig. Layer-specific quantification of RS and FS neuronal activity in ALM. (A)** Delay activity in ALM RS neurons upon learning. Left: baseline-subtracted (1 second prior to whisker onset) population firing rate (mean ± SEM) for different cortical layers (L2/3, L5, and L6a) overlaid for Novice mice (234 neurons in 6 mice) and Expert mice (766 neurons in 12 mice). The number of neurons for each layer is indicated on the figure. Right: change in average spike rate quantified in 200- to 1,000-ms window after whisker onset relative to similar window size before whisker onset. ***: $p < 0.001$, **: $p < 0.01$, ns: $p > = 0.05$, nonparametric permutation test, FDR-corrected for multiple comparison. **(B)** Delay activity in ALM FS neurons upon learning. Panels are similar to (A) but for ALM FS neurons in Novice mice (37 neurons in 5 mice) and Expert mice (109 neurons in 12 mice). **(C)** Fraction of ALM RS neurons across different layers with significant positive (filled bars) or negative (empty bars) modulation during delay period (200- to 1,000-ms window after whisker onset relative to similar window size before whisker onset). Positive or negative modulation of neurons was quantified using nonparametric permutation test ($p < 0.005$). ***: $p < 0.001$, *: $p < 0.05$, ns: $p > = 0.05$, chi-squared proportion test. Fractions are reported for groups with more than 5 neurons. **(D)** Similar to (C) but for ALM FS neurons. **(E)** Mouse-by-mouse variability and distribution of delay activity of RS units in ALM of Novice and Expert mice. (Left) Bar plots showing average firing rate across mice in 200- to 1,000-ms window (mean ± SEM, 6 Novice and 12 Expert mice) after whisker onset and statistical comparison using nonparametric permutation test (*: $p < 0.05$). Circles show individual mice. (Right) Violin plots showing the distribution of delay activity in 200- to 1,000-ms window for all neurons recorded in Novice (234 neurons in 6 mice) and Expert mice (766 neurons in 12 mice). **(F)** Same as (E) but for ALM FS units in Novice (37 neurons in 5 mice) and Expert mice (109 neurons in 12 mice). The underlying data for S15 Fig can be found in S9 Data. ALM, anterior lateral motor cortex; FDR, false discovery rate; FS, fast spiking; RS, regular spiking.
(TIF)

**S16 Fig. Preparatory neuronal activity in ALM is decreased, but remains significant in quiet trials devoid of movements. (A)** Larger delay period activity of ALM RS neurons in Active versus Quiet hit trials. Left: baseline-subtracted (1 second prior to whisker onset) population firing rate (mean ± SEM) overlaid for Quiet (blue) and Active (red) hit trials in Expert mice (766 RS units in 12 mice). Right: change in average spike rate quantified in 200- to

1,000-ms window after whisker onset relative to a 1-second window prior to whisker onset. ***: $p < 0.001$, nonparametric permutation test. Asterisks below the bars represent the $p$-value for comparing delay activity in each trial type compared to baseline, while asterisks above the bars represent the $p$-value of the comparison between delay activity in Quiet and Active hit trials. **(B)** Similar to (A) but for FS neurons (109 FS units in 12 mice). **(C)** Similar to (A) but separately for RS neurons of different cortical layers. The number of neurons for each layer is indicated on the figure. ***: $p < 0.001$, ns: $p > = 0.05$, nonparametric permutation test, FDR-corrected for multiple comparison. **(D)** Similar to (C) but for FS neurons. The underlying data for S16 Fig can be found in S9 Data. ALM, anterior lateral motor cortex; FDR, false discovery rate; FS, fast spiking; RS, regular spiking.
(TIF)

**S1 Data. Data underlying Figs 1, S2, S3, and S4.**
(XLSX)

**S2 Data. Data underlying Fig 2.**
(XLSB)

**S3 Data. Data underlying S5 Fig.**
(XLSB)

**S4 Data. Data from Novice mice underlying Figs 3, S6, and S7.**
(XLSB)

**S5 Data. Data from Expert mice underlying Figs 3, S6, and S7.**
(XLSB)

**S6 Data. Data underlying Figs 4, S8, and S9.**
(XLSX)

**S7 Data. Data underlying Fig 5.**
(XLSX)

**S8 Data. Data underlying Figs 6, S10, S11, and S12.**
(XLSX)

**S9 Data. Data underlying Figs 7, S13, S14, S15, and S16.**
(XLSX)

**S10 Data. Data underlying Fig 8.**
(XLSX)

## Author Contributions

**Conceptualization:** Vahid Esmaeili, Keita Tamura, Sylvain Crochet, Carl C. H. Petersen.

**Data curation:** Vahid Esmaeili.

**Formal analysis:** Vahid Esmaeili, Anastasiia Oryshchuk, Reza Asri, Romain Guiet.

**Funding acquisition:** Keita Tamura, Carl C. H. Petersen.

**Investigation:** Vahid Esmaeili, Anastasiia Oryshchuk, Georgios Foustoukos, Yanqi Liu.

**Methodology:** Vahid Esmaeili, Anastasiia Oryshchuk, Keita Tamura, Georgios Foustoukos, Yanqi Liu, Romain Guiet.

**Software:** Vahid Esmaeili.

**Supervision:** Sylvain Crochet, Carl C. H. Petersen.

**Visualization:** Vahid Esmaeili, Romain Guiet.

**Writing – original draft:** Vahid Esmaeili, Keita Tamura, Sylvain Crochet, Carl C. H. Petersen.

**Writing – review & editing:** Vahid Esmaeili, Anastasiia Oryshchuk, Reza Asri, Keita Tamura, Georgios Foustoukos, Yanqi Liu, Romain Guiet, Sylvain Crochet, Carl C. H. Petersen.

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
