## [Editor Report · Decision Letter 0]

21 Sep 2021

Dear Carl, 

Thank you for submitting your manuscript entitled "Learning-related congruent and incongruent changes of excitation and inhibition in distinct cortical regions" for consideration as a Research Article by PLOS Biology.

Your manuscript has now been evaluated by the PLOS Biology editorial staff, as well as by an academic editor with relevant expertise, and I am writing to let you know that we would like to send your submission out for external peer review.

Please re-submit your manuscript within two working days, i.e. by Sep 23 2021 11:59PM.

Kind regards,

Gabriel Gasque

Senior Editor

PLOS Biology

ggasque@plos.org

---

## [Decision Letter · Decision Letter 1]

20 Oct 2021

Dear Carl,

Thank you for submitting your manuscript "Learning-related congruent and incongruent changes of excitation and inhibition in distinct cortical regions" for consideration as a Research Article at PLOS Biology. Your manuscript has been evaluated by the PLOS Biology editors, by an Academic Editor with relevant expertise, and by four independent reviewers.

In light of the reviews (below), we will not be able to accept the current version of the manuscript, but we would welcome re-submission of a much-revised version that takes into account the reviewers' comments. We cannot make any decision about publication until we have seen the revised manuscript and your response to the reviewers' comments. Your revised manuscript is also likely to be sent for further evaluation by the reviewers.

We expect to receive your revised manuscript within 3 months. 

**IMPORTANT - SUBMITTING YOUR REVISION**

Your revisions should address the specific points made by each reviewer. As you will see, the reviewers are overall supportive, but raise a series of major and minor concerns that should be addressed for us to consider a revision. Having discussed these comments with the Academic Editor, we encourage you to consider the analyses suggested by reviewer 3 in her/his major point. If these analyses are not possible, you should explain why. You should also add to your manuscript the controls requested by reviewer 1 (point 2) or clearly cite previous work that would indicate those controls are not needed at this point. We have determined that the additional experiments recommended by reviewer 1 (whole cell recordings) are beyond the scope of the current study. Thus, you don’t need to add those data. 

Please submit the following files along with your revised manuscript:

*Re-submission Checklist*

*Published Peer Review*

*PLOS Data Policy*

*Blot and Gel Data Policy*

Sincerely,

Gabriel Gasque

Senior Editor

PLOS Biology

ggasque@plos.org

REVIEWS:

Reviewer #1: Training rodents to report sensory percepts by licking a port for reward is a very commonly used experimental paradigm for investigating sensorimotor transformations, but knowledge about the differential roles played by excitatory and inhibitory neurons in such tasks is surprisingly lacking. The authors attempted to address this gap by examining the activity of putative excitatory (RS) and inhibitory (FS) neurons separately in a whisker-based delayed response task. They found that FS cells are strongly activated in all the cortical regions studied. In most cortical areas, the process of learning the task led to changes in FS cell activity that were similar to changes in RS cell activity. Interestingly, however, in two regions, wM1 and wM2, FS cell and RS cell activity changed in opposite directions due to learning. This finding also corroborates some observations from their previous paper investigating the same task.

Overall, the paper is of good quality and describes significant results that help fill important gaps in knowledge, and that are of general interest in systems neuroscience. Most of the claims made in the manuscript are backed up by the evidence presented, and the statistical analyses are satisfactory. My main complaints are that the authors could have taken some of the interesting findings a lot further.

Major points

1. The learning-related changes in activity of RS and FS cells in opposite directions in wM1 and wM2 were probably the most interesting and novel observations in the paper, but the authors did not pursue this avenue of inquiry fully. For example, what are the circuit-level mechanisms that bring these changes about? To what extent are they dependent on plasticity at synapses onto FS and RS neurons? Are the changes in the excitation/inhibition ratio at all important for the animal to perform the task? Whole cell recordings from identified cell types could help address some of these questions.

Several of the earlier figures summarize time-courses of FS and RS cell activity, but these findings don't seem to be particularly novel or unexpected. As the authors themselves acknowledge, FS cell activity patterns generally mirror RS cell activity, but with shorter latencies and larger amplitudes. While these are valuable results that should be reported, it is perhaps not surprising that this should be the case given that FS cells are well-known to form powerful feed-forward inhibitory connections within and between cortical regions. Instead, devoting more space to fully exploring the circuit-level processes underlying the changes in the excitation/inhibition balance in wM1 and wM2 would go a long way towards improving the manuscript.

2. Concerning Fig 4 E, the authors' claim of a differential effect on licking due to optogenetic inactivation of wS1 vs wS2 would be interesting, but this is not sufficiently backed up by the data. No controls for the optogenetic stimulation parameters were included - wS2 encompasses a smaller region of cortical surface than wS1, and so one might expect the same optical stimulation parameters to inhibit a much larger proportion of S2 neurons, thereby explaining the observed behavioral effect. Including some assays with different levels of optical stimulation intensities, and control experiments with stimulation of cortical areas unrelated to the task would help to make this part a lot more convincing. 

Minor points

1. The second paragraph of the Introduction seems to be somewhat broad. It would be helpful to add some more details about known roles of FS cells in cortex-dependent learning in order to help contextualize the results in the literature.

2. It is not clear from the text whether Novice and Expert mice had overlapping sets of animals.

3. Figure 4 C-D have missing x-axis labels.

Reviewer #2: A major goal in neuroscience is to identify the neuronal circuits through which sensation is transformed into action. Esmaeili et al. has previously characterized how neurons along the sensori-motor hierarchy respond to stimuli as mice learn to report detection of whisker input. The current study reanalyzes this previous dataset to identify fast-spiking (FS; putative interneurons) and regular spiking (RS; putative pyramidal) cells and compare their strength and timing of recruitment across learning. The authors validate their classification of FS and RS cells using optotagging and comparisons of firing rates, and also retain appropriate caution in the assignment of cell types. Interestingly, they find that RS and FS cells change largely in parallel in sensory areas (wS1 and wS2) and in regions associated with motor output and licking (ALM and tjM1), but have opposite effects in whisker related motor areas (wM1 and wM2). Namely, RS neurons increase their activity relative to FS in wM2, but decrease their activity relative to FS neurons in wM1. This area- and cell-type specific plasticity is an exciting and novel example of how learning impacts cortical computations and will be of interest to a range of investigators studying circuits and behavior across systems and species. Overall, the study is well written and clearly presented (an accomplishment for a study with so many conditions) and I only have a few suggestions to improve robustness of the study.

1. The most important claim for the paper is the ability to use spike waveforms to identify putative inhibitory interneurons.

a. The optotagging approach is an important control to understand the potential incidence of false positives and false negatives. Thus, it would be preferable to include these data showing the relationship between spike width and ChR2 response in the main figures rather than the supplement. However, the methods provide limited information for how this data was collected and analyzed. Stimulus conditions are provided for the optrode recordings (although these are confusingly discussed in the context of wS1 inactivation) but not the LED-mediated activation, and no information is given for the analysis window that was used to measure firing rates. At least in the case of the optrode recordings, the stimulus window is long (600 ms) and therefore it might not be ideal to measure firing during the whole window as a proxy for ChR2 expression due to network effects (see for example Sanzeni et al., eLife, 2020).

b. As shown in Figure S2, the relative fraction of FS to RS cells varies substantially across areas. It is surprising that more than half of the cells in wS1 are characterized as FS (though notably this is only true for the behavioral dataset; the optotagging dataset is much more biased towards RS cells). Some of this likely reflects true differences in the density of these cell types across areas, but also likely reveals some sampling biases. The authors should provide some discussion for how such a bias might impact their results.

c. Given that the authors have simultaneous recordings from both FS and RS cells, it would be worthwhile to test whether these identities could be corroborated by functional measures, for instance testing connectivity using cross-correlations. Clearly, this would not be a high-yield approach, but could provide some internal validation of their behavior dataset.

2. The authors provide some interpretation of the increase in relative excitation in wM2 as increasing the drive to downstream motor areas such as ALM, but there is little discussion for what the role of the decrease in excitation of wM1 might mean for the circuit or behavior. 

3. All of the data provided in the study are the averages of many neurons from multiple animals. The main figures should provide a better sense of the variability/robustness in the dataset: e.g. the distribution of firing rates across the population and/or comparisons of FS and RS cells on a mouse-by-mouse basis. 

4. It is not clear if the optogenetic inactivation is new data in this manuscript. 

5. The x-axis of the time-courses are in trial time- it would be useful to have additional cues to remember what the relevant events are (e.g. whisker stim) in each figure.

Reviewer #3: The work of Esmaeili and colleagues reports the learning-related changes in whisker sensory, whisker motor and orofacial motor areas. This work builds on findings reported in a recent publication from Petersen lab. They further analyzed the data set of single unit recordings made simultaneously in six cortical regions by resolving changes in firing rates of specific neuronal subtypes - putative excitatory and inhibitory neurons. Main findings of the work are following. Little learning-related changes were observed in whisker primary and secondary somatosensory areas (wS1, wS2), although microcircuit-level changes could not be resolved in the study. In whisker primary and secondary motor areas (wM1 and wM2), excitatory and inhibitory neurons changed whisker-evoked responses in opposite directions. In wM1, inhibitory neurons increased firing rate whereas excitatory neurons decreased firing rate across learning. In wM2, inhibitory neurons decreased firing rate while excitatory neurons increased firing rate. In orofacial areas, excitatory and inhibitory neurons showed congruent changes because both subtypes either increased (in ALM) or decreased (in tjM1) the whisker evoked activity. This is an important analysis of data set acquired in a well-conducted study. The text is concise and generally clear, figures are of a good quality. However, I have several comments and suggestions. 

Major points:

In its present form, the present study is focused on describing learning-related changes of excitation and inhibition in key cortical regions involved in the task but falls short of providing mechanistic insights into how changes in one area are related to those in another area. Figure 5 shows anatomical data suggesting that learning-related changes in wM2 are driven by wS2. If this is the case, activities in wS2 and wM2 would show enhanced coordination in expert mice. Establishing a causal relationship between wS2 and wM2 ultimately requires manipulation of wS2 while recording from wM2, which would be beyond the scope of the study. Nonetheless, one could use 'spike-triggered' approach as described in 2015 Neuron paper by Zandvakili and Adam Kohn where they showed V2 activity was preceded by increased coordination in V1 activity. Alternatively, one could infer a causal relationship using granger causality as described in the Neuron paper from Patrick Kanold's lab (Francis et al., 2018). Without providing additional information on functional changes in coordinated activity across cortical regions, I feel that the authors are not making full use of the excellent data set they collected and that Figure 5 is disjointed from the rest of the manuscript. 

Minor points:

1. RS and FS units in wS1: Supp. Fig. 3 shows no significant difference in AP rates of RS and FS units in wS1. This is a bit puzzling to me because FS units are known to show higher firing rate. Consistent with this, Supp. Fig. 4 and Fig. 1L show elevated firing rate of FS units before the onset of photostimulation. This does not impact the conclusion of the paper, but is worth clarifying. 

2. On page 7, paragraph 1, it is stated that "fast dynamic changes in the relative timing of the recruitment of FS and RS units across cortical regions". The statement implies that the changes occur across cortical regions, although they only occur in wM1 and wM2 (Figure 3). The statement could be toned down a bit.

3. Figure 2: a plot showing first lick probability in novice and expert animals could be included and aligned to the firing rate traces. 

4. Figure 4 focuses on fast whisker-evoked responses (50 ms after the stimulus onset). No significant changes were observed in the response of RS or FS units in wS1 and wS2 during this short time window. Petersen lab previously reported the presence of 'late' depolarization or excitation that emerges learning-dependent manner. These 'late' responses are clearly shown in average traces of FS and RS activities (Figure 2) but were not analyzed. Could the analysis of AP rate change (Fig. 4C) be extended to the 'late' component of the response for both wS1 and wS2? 

5. Figure 5: do any of S2M2 projections arrive in ALM or M1? Suter and Shepherd 2015 showed that wS2 and wM1 are reciprocally connected, so I would expect some projections in wM1. Quantification of projection fields would be helpful to address these points. 

6. Supp. Fig. 13: there is a typo in the heading. "RS and FS neuronal activity in tjM1"  "....in ALM"

Reviewer #4, Michael Brecht: In their paper ‚Learning-related congruent and incongruent changes of excitation and inhibition in distinct cortical regions' Esmaeili et al. describes changes in fast spiking and regular spiking neurons during perceptual learning across a set of cortical areas in the mouse. This study ties in with a large body of work that the authors performed around similar tasks in mice. The experiments are performed to a high experimental standard, there appear to be no animal welfare issues. The key finding is that fast spiking and regular spiking neurons do not show the same learning related changes across cortical areas. I do think this is a potentially important result. Most of the other findings are minor. I think it would be great if the authors would provide a bit more background on their key finding, otherwise I think this study suitable for publication.

Major

Are the interneurons the same across the cortical regions investigated?

It would be great if the authors couzld add info on the laminar distribution of their effects.

What is the author's idea on the functional significance of their finding?

Minor

Title: are we talking about cortical regions or cortical areas?

Figure 2: The annotation hit is slightly confusing; replace with 'hit trials' or 'hit trials only'?

---

## [Decision Letter · Decision Letter 2]

19 Apr 2022

Dear Carl,

Thank you for submitting your revised Research Article entitled "Learning-related congruent and incongruent changes of excitation and inhibition in distinct cortical areas" for publication in PLOS Biology. I have now obtained advice from the original reviewers and have discussed their comments with the Academic Editor. 

Based on the reviews, we will probably accept this manuscript for publication, provided you satisfactorily address the remaining points raised by the reviewers. 

You will see that, in the final round, Reviewer 2 raised one final concern - suggesting "that these measures are extremely sensitive to overall firing rates, which also change across learning, and the authors have not performed the necessary control analyses to ensure that this can not explain the observed effects." Upon consulting with our Academic Editor and the other Reviewers a few suggestions have been made regarding how you could address this minor but relevant concern. Reviewer 3 suggested a variety of approaches which have been appended below with the additional reviewer feedback. The Academic Editor also suggested that you could measure pairwise correlations using the spike time tiling coefficient (or STTC) using the data you've already acquired - suggesting that STTC is independent of differences in firing rates that indeed can confound estimates of correlated activity (Cutts and Eglen, J. Neurosci. 2014; PMID: 25339742). The referenced paper describes this method in detail. By addressing this remaining issue, we feel that you will have a better controlled estimate of correlated activity in the presence of different firing rates.

We'd also like you to consider a title change that removes "congruent" and "incongruent" if you are amenable as we feel those terms are not as broadly accessible as we'd like. We've come up with two suggestions, but please feel free to use an alternative if you aren't keen on either. The second is, admittedly, rather long and may be an overstatement.

1) Different learning-related changes in excitation and inhibition across regions of the sensorimotor cortex

2) Different learning-related changes in excitation and inhibition across regions of the sensorimotor cortex contribute to delays in transforming sensory input to motor responses

Please also make sure to address the following data and other policy-related requests. These are required to be addressed in full before we can move to final acceptance.

We hope to receive your revised manuscript within two weeks. If, however, you feel that you will need additional time to address Reviewer 2's concerns, please reach out to let me know.

*Published Peer Review History*

*Press*

Please do not hesitate to contact me should you have any questions. I apologize again for how long it took to convey this decision to you.

Sincerely,

Kris

Kris Dickson,

Neurosciences Senior Editor/Section Manager,

kdickson@plos.org,

PLOS Biology

ETHICS STATEMENT: While you've indicated an ethics statement is included in the methods section of the paper, I did not see it. I apologize if I missed it. But if not, please make sure to add it to the study. This needs to include the NAME of the ethics committee and the license and approval NUMBER, or needs to be noted that you've received “verbal” approval.

-- Please include the full name of the IACUC/ethics committee that reviewed and approved the animal care and use protocol/permit/project license. Please also include an approval number.

-- Please include the specific national or international regulations/guidelines to which your animal care and use protocol adhered. Please note that institutional or accreditation organization guidelines (such as AAALAC) do not meet this requirement.

-- Please include information about the form of consent (written/oral) given for research involving human participants. All research involving human participants must have been approved by the authors' Institutional Review Board (IRB) or an equivalent committee, and all clinical investigation must have been conducted according to the principles expressed in the Declaration of Helsinki.

DATA POLICY:

2) Deposition in a publicly available repository. Please also provide the accession code or a reviewer link so that we may view your data before publication. ***I appreciate that you've indicated that the DOI will be made available at Open Access CERN database Zenodo: https://zenodo.org/communities/petersen-lab-data **with doi hyperlink**. Please note that we must have this information accessible prior to final acceptance so that it can be transferred to production upon acceptance. 

Regardless of the method selected, please ensure that you provide the individual numerical values that underlie the summary data displayed in the figure panels as they are essential for readers to assess your analysis and to reproduce it. This impacts most of your figure panels. I've listed them below, but please double check that I didn't miss any.

Figures 1G-O; 2B&D; 3A,C,D; 4A-E; 6A-F; 7A-D; 8C,E

Supplementary Figures: 1E; 2A,B; 3A-F, 4A-F, 5A-D. 6A-B, 8A-D, 9A-H, 10A-F, 11A-F; 12AB; 13A-F, 14A-F; 15A-D; 16A-D

IMPORTANT: Please also ensure that figure legends in your manuscript include information on **where the underlying data can be found, and ensure your supplemental data file/s has a legend.** This usually takes the form of statements at the end of each legend saying "the raw data for Fig X can be found at Y." This statement needs to be included for all relevant main and supplemental figures.

DATA NOT SHOWN?

Reviewer remarks:

Reviewer's Responses to Questions

PLOS authors have the option to publish the peer review history of their article (what does this mean?). If published, this will include your full peer review and any attached files.

Reviewer #1: No

Reviewer #2: No

Reviewer #3: No

Reviewer #4: Yes: Michael Brecht

Reviewer #1: This version of the manuscript has been greatly improved from the first one. The new analyses of trial-by-trial firing rate covariances provide valuable insights into the potential underlying mechanisms of the differential region-specific effects of learning, and constitute an extremely useful addition to the paper. I find that my other concerns have also been addressed satisfactorily and as such, I have no new ones to add.

Reviewer #2: Esmaeili et al. have responded thoroughly to the concerns put forth by the reviewers and have substantially increased the clarity, rigor and insight of the manuscript. The new correlation analyses are particularly exciting in their potential to provide mechanistic insights into the changes in connectivity across areas that occur with learning. My only concern is that these measures are extremely sensitive to overall firing rates, which also change across learning, and the authors have not performed the necessary control analyses to ensure that this can not explain the observed effects. 

Reviewer #3: All of my concerns have been sufficiently addressed. The manuscript contains data of high-quality and offers important insights.

Additional suggestions from Reviewer 3:

Reviewer #2's concern is a fair one, but I think the authors partly mitigated it. The reviewer seems to suggest that for wS2wM2 pathway, there is an increase in the firing rate of RS units in wM2 (Fig. 6C), which can lead to the apparent increase in the cross-correlation wS2wM2 RS (Fig. 6F, third panel) as a secondary effect. As far as I can see, there are a couple of things that authors could do to address it. I actually feel that the authors might already have partly addressed it with the data in the manuscript.

For example, an argument against the reviewer's point can be found in wS1wM1 FS (Fig. 6F, second panel). Note that wM1 FS units showed roughly 2-fold increase in firing during learning (Fig. 6B), yet wS1wM1 FS cross-correlation did not significantly increase (Fig. 6F, second panel). In general, increases in cross-correlation are not likely to be a byproduct of increases in firing rate. But of course, here we are talking about two different pathways wS2wM2 vs wS1wM1 and two different cell-types (RS vs FS). The fact that learning-induced increases in firing rate do not contribute to changes in correlation may not be generalized across different pathways.

Another way would be to test increases in cross-correlation between wS2 and another brain region as a 'control'. This brain region should also show increases in firing rate, for it to be a fair comparison. ALM fits this criterion. In Fig. 7C, the firing rate of ALM RS units increases. Maybe the authors could look at wS2ALM RS cross-correlation and see if it increases to a similar extent to wS2wM2 RS?

Another way would be to use simulation. One could create 2-level neural network (wS2 and wM2) before and after learning that consists of RS units that fire spikes with known statistics (poisson distribution). Based on the observed increases in RS firing rate in wM2, one could then simulate the learning-induced increases in RS firing rate in wM2 and calculate cross-correlation. Compare it against 'naive' condition and test if there is an increase in cross-correlation.

Reviewer #4: The authors have addressed my concerns. More than that they also addressed the quite comprehensive remarks of the other referees. With all that I support publication.

Minor

Could Figure 1 be perhaps be a bit condensed or be split into two Figures? Going up to Panel O (with even more subpanels present) feels unhealthy; the result will be that anyhow few readers will work through this much detail.

---

## [Editor Report · Decision Letter 3]

10 May 2022

Dear Carl,

On behalf of my colleagues and the Academic Editor, Alberto Bacci, I am pleased to say that we can in principle accept your Research Article "Learning-related congruent and incongruent changes of excitation and inhibition in distinct cortical areas" for publication in PLOS Biology. Alberto looked over the new analyses you'd provided and was satisfied with them. and with your other responses to the remaining reviewer feedback. We are also fine with you keeping your original title given, as you note, that congruent and incongruent convey specific information that is important to the conclusions you draw. It is also fine to keep Figure 1 as it is. I also apologize for missing the Ethics statement the last time I read through your methods section. What you provided is exactly what we need.

At this stage, we simply need you to address any remaining formatting and reporting issues. These will be detailed in an email from our production team that will follow this letter and that you will usually receive within 2-3 business days, during which time no action is required from you. Please note that we will not be able to formally accept your manuscript and schedule it for publication until you have completed any requested changes.

PRESS

Sincerely, 

Kris

Kris Dickson, Ph.D. 

Neurosciences Senior Editor/Section Manager 

PLOS Biology

kdickson@plos.org